

# INCHEM-Py v1.2: A community box model for indoor air chemistry

David R. Shaw[1], Toby J. Carter[1], Helen L. Davies[1], Ellen Harding-Smith[1,2], Elliott C. Crocker[3], Georgia Beel[1,4], Zixu Wang[1], and Nicola Carslaw[1]

[1]Department of Environment and Geography, University of York, York, YO10 5NG, United Kingdom
[2]Wolfson Atmospheric Chemistry Laboratories, University of York, York, YO10 5DD, United Kingdom
[3]Department of Chemistry, University of York, York, YO10 5DD, United Kingdom
[4]UK Centre for Ecology and Hydrology, Edinburgh, EH26 0QB, United Kingdom

**Correspondence:** David R. Shaw (david.shaw@york.ac.uk)

**Abstract.** The Indoor CHEMical model in Python, INCHEM-Py, is an open-source and accessible box model for the simulation of the indoor atmosphere, and is a refactor and significant development of the INdoor Detailed Chemical Model (INDCM). INCHEM-Py creates and solves a system of coupled ordinary differential equations that include gas-phase chemistry, surface deposition, indoor/outdoor air change, indoor photolysis processes and gas-to-particle partitioning for three common terpenes. It is optimised for ease of installation and simple modification for inexperienced users, while also providing unfettered access to customise the physical and chemical processes for more advanced users. A detailed user manual is included with the model and updated with each version release. In this paper, INCHEM-Py v1.2 is introduced, the modelled processes are described in detail, with benchmarking between simulated data and published experimental results presented, alongside discussion of the parameters and assumptions used. It is shown that INCHEM-Py achieves excellent agreement with measurements from two experimental campaigns which investigate the effects of people and different surfaces on the concentrations of different indoor air pollutants. In addition, INCHEM-Py shows closer agreement to experimental data than INDCM. This is due to the increased functionality of INCHEM-Py to model additional processes, such as deposition-induced surface emissions. Published community use-cases of INCHEM-Py are also presented to show the variety of applications for which this model is valuable to further our understanding of indoor air chemistry.

## 1 Introduction

In recent years, the quality of the air we breathe has gained increased attention. The World Health Organisation (WHO) recently stated that exposure to indoor and outdoor air pollution was one of the greatest risks to human health, and that improving air quality was necessary to reduce the global incidence and impact of diseases such as lung cancer, stroke and asthma (World Health Organisation, 2001). Much of the focus in developed countries to date has been on outdoor air quality, particularly in relation to attaining regulatory guidelines. However, exposure to air pollution indoors is arguably more important, given we are estimated to spend around 90% of our time indoors in high income countries like the United Kingdom (Klepeis et al., 2001). Moreover, the recent COVID-19 pandemic has highlighted the importance of good indoor air quality, particularly around virus transmission.





Indoor air is a complex mixture of gas and particles, primarily as a result of numerous emission sources. These include
activities such as cooking, cleaning, air freshener or scented candle use, household improvement activities (such as painting),
emissions from building and furnishing materials (Uhde and Salthammer, 2007; Weschler, 2009), and from building occupants
through breathing and interactions with compounds in skin (Nazaroff and Weschler, 2004; Wisthaler and Weschler, 2010).
Typical indoor pollutants include Volatile Organic Compounds (VOCs), nitrogen oxides ($NO_X$) and particulate matter (PM).
In addition, VOCs produced from indoor sources, or via ingress into buildings through windows doors and cracks in the
building fabric, can evolve over time through reactions with oxidant species, such as ozone ($O_3$) (Liu et al., 2021), hydroxyl
radicals (OH) (Carslaw et al., 2017) and nitrate radicals ($NO_3$) (Arata et al., 2018). This secondary chemistry can result in
the formation of a wide range of secondary products, such as formaldehyde and secondary organic aerosol (SOA), which can
have a high potential for causing health issues, when compared to the parent VOC. For example, formaldehyde is a known
carcinogen, and SOA are ultra-fine particles, which are associated with a range of cardiovascular and respiratory conditions
(Schraufnagel, 2020; WHO Air quality and health Guidelines Review Committee, 2010). Given that long-term monitoring
of such processes has been limited to date, indoor air chemistry models provide a useful tool to aid further understanding,
particularly of the evolution of secondary chemistry over time.

In recent years, there have been significant advances in our understanding of indoor air quality. This is due, in part, to
some larger-scale, intensive indoor measurement campaigns in test-houses, involving a suite of advanced instrumentation more
typically used in outdoor field campaigns (see the overview in Farmer et al. (2019)). The results from these campaigns have
demonstrated that indoor air quality is a complex, multidimensional problem, where indoor and outdoor sources, transformation
processes, and building design, management and use are the driving physical and chemical factors for air quality. Human
behaviour then adds further complexity. Ideally, one would perform numerous measurements in numerous buildings and for
long periods of time to understand these processes. However, such a task would be expensive, time consuming and logistically
challenging. For instance, how would representative buildings be selected? An alternate approach is to simulate these processes
in a model. To achieve this goal requires a thorough understanding of the fundamental processes that underpin indoor air quality
in buildings.

There are numerous processes that need to be considered in a model of indoor air quality, these include direct and secondary
emissions from surfaces (building materials, furnishings and people), physical building parameters (such as ventilation rates,
temperature, humidity and light), gas- and particle-phase chemistry, surface deposition, and the effect of occupants. These
factors will combine to determine the occupant exposure to air pollution under any set of building conditions. Note that
microbial emissions also play a role in occupant exposure to indoor air pollution, but they have not been considered in this
work.

The INdoor CHEMical Model in Python (INCHEM-Py) is a zero dimensional community box model that includes many of
these processes to enable investigation of the evolution of indoor air chemistry over time. It is a refactor and improvement of
the INdoor Detailed Chemical Model (INDCM) developed by Carslaw (2007). INCHEM-Py v1.1 has been described in basic
detail, covering accessibility and broad function in Shaw and Carslaw (2021). This paper describes the model in more detail,




including the developments included in the latest version (v1.2), and gives some examples of the ways in which it can be used by the scientific community.

## 2 Model description


INCHEM-Py creates and solves a series of first order Ordinary Differential Equations (ODEs), with each equation representing the rate of change of a species concentration (molecule $cm^{-3}$) with time. The general equation for these ODEs is shown for the concentration (C) of species $i$ in Eq. (1).

$$\frac{dC_i}{dt} = \sum R_{ij} + (\lambda_r C_{i,out} - \lambda_r C_i) - \nu_d \left(\frac{A}{V}\right) C_i \pm k_t C_i \qquad (1)$$

The chemical mechanisms are represented with the first term on the right hand side of Eq. (1) as a sum of all reaction rates involving species $i$, with $j$ representing other species. Each reaction between species takes place at a rate, $R$, dependent on the concentrations of the reactant species and the rate constant. The second term of each ODE accounts for indoor/outdoor exchange, which depends on the air change rate, $\lambda_r$, and the species concentrations both indoors ($C_i$) and outdoors ($C_{i,out}$). The third accounts for surface deposition, dependent on the species deposition velocity, $\nu_d$, and surface to volume ratio (A/V),
while the final term is for timed emissions, where $k_t$ is the emission/loss rate of species $i$ at time $t$.

Rate coefficients and reactions are imported in the most part from the Master Chemical Mechanism (MCM) (Jenkin et al., 1997, 2003, 2015, 2018, 2020; Saunders et al., 2003; Bloss et al., 2005), an almost explicit chemical mechanism following the atmospheric degradation of 135 VOCs. Additional schemes and mechanisms for species that are not included in the MCM, but which do occur indoors, are parsed from an additional file. Schemes have been developed for some terpenoids that are present
in many scented items found and used indoors (Carslaw, 2007, 2013; Carslaw et al., 2012, 2017; Terry et al., 2014). Chlorine schemes have also been designed and implemented, since it is a key pollutant from bleach cleaning products (Wong et al., 2017). The full list of additional schemes is available in the user manual. The model also includes the function for users to add their own chemical mechanisms to supplement those already included, thereby allowing reaction mechanism development, and community-driven model development.

All species within the model are assumed to egress with a specified air change rate ($\lambda_r$) as a function of their indoor concentration ($C_i$). Some species also ingress at the same air change rate, depending on their constant or diurnal outdoor concentrations ($C_{i,out}$). Species are also assumed to irreversibly deposit to surfaces at a rate dependent on individual deposition velocities ($\nu_d$, mostly taken from Carslaw et al. (2012)) and the surface area to volume ratio (A/V) of the simulated space, as in Carslaw (2007). An optional mechanism for $O_3$ and $H_2O_2$ surface deposition and subsequent emission has been developed with
a detailed discussion in Sect. 2.8. Some emissions from surfaces are included in the new surface mechanisms, but additional emissions, both constant and intermittent, can be user defined. The maximum size of the chemical mechanism solved by INCHEM-Py (v1.2), before any additional user mechanisms are added, is 6507 chemical species and particles undergoing 19581 reactions.





## 2.1 ODE formation and solution

All gas phase reactions solved in INCHEM-Py are of the general form

$$C_i + ... + C_j \overset{k}{=} C_x + ... + C_y \tag{2}$$

where $C_i$ remains the concentration of an individual species and $k$ is the rate constant of the reaction, which may be constant, or depend on other variables such as photolysis. In each case, the concentration of species on the right hand side of the equation are increased and the species on the left hand side (lhs) of the equation are decreased at the same rate ($R$) of

$$R_{ij} = \sum C_{lhs} k \tag{3}$$

Therefore, by calculating $R$ for each species in each reaction, the total change in an individual species concentration only through gas-phase reactions is given by the sum of all of the individual reaction rates, as follows:

$$\frac{dC_{i,Gas}}{dt} = \sum R_{ij} \tag{4}$$

Particles are discussed in Sect. 2.4, but the partitioning equations follow the same form as Eq. (2) and are, therefore, imple-
mented into the ODEs in the same way, with an ODE for each particle concentration.

The ODEs for each species are solved simultaneously using integrate.ode from the Scipy Python library and the wrapped LSODA function from the Fortran solver package ODEPACK (Hindmarsh, 1983). LSODA was chosen as the solution method as it automatically switches between the implicit Adams-Moulton formulae for non-stiff problems and Backward Differentiation Formulae (BDF) for stiff problems. Due to the size of the system, and its highly-coupled structure, the problem is
stiff in most cases (note that if using only a small selection of the MCM mechanism it may be non-stiff in some instances). LSODA makes use of the Jacobian of the system to ease concentration predictions and is calculated in full as an array of partial differential equations. Representing Eq. (1) as $f(t, C_i)$ the (i,j)th entry in the Jacobian is

$$J_{ij} = \frac{\partial f_i}{\partial C_{S_j}} \tag{5}$$

## 2.2 Photolysis

Time is given in local solar time in INCHEM-Py and determined using the date and latitude of the simulation. A date input is used to calculate the solar declination angle (Dec) as

$$Dec = -23.45 \times \cos\left(\frac{360}{365.25} \times (d + 10)\right) \tag{6}$$

where $d$ is the number of days since the 1st of January that year. Figure 1 shows the relationship between the declination and the solar zenith angle, $\theta$, which is used to calculate the outdoor photolysis rates. In Figure 1, CS represents a line connecting
the centre of the Earth (C) with the centre of the Sun, Z represents the zenith of the location of the simulation, Lat is the latitude of the location of the simulation, and LHA is the local hour angle which is 0 at solar noon. Using these known values and the spherical law of cosines the solar zenith angle is calculated as

$$\cos(\theta) = \sin(Lat)\sin(Dec) + \cos(Lat)\cos(Dec)\cos(LHA) \tag{7}$$



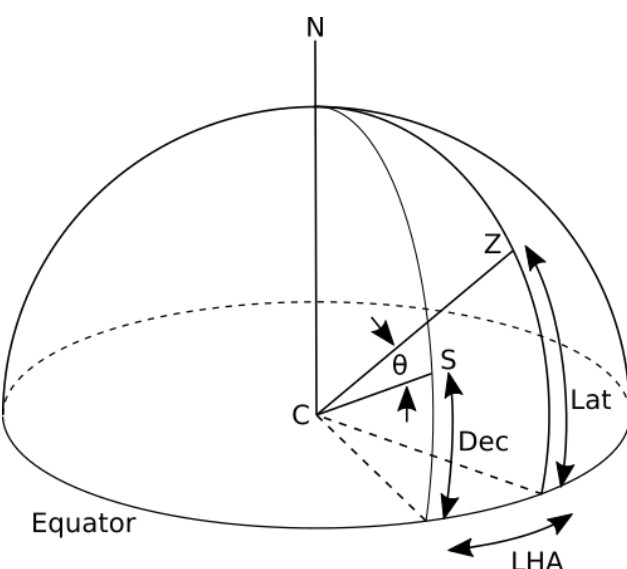

**Figure 1.** Relationship between declination angle (Dec), latitude (Lat) and solar zenith angle ($\theta$). C is the centre of the Earth, the line CS is a line connecting the centre of the Earth to the centre of the Sun and Z represents the zenith of the location being simulated.

The solar zenith angle is used to calculate the solar photolysis portion of the overall photolysis rate ($J$) for the 43 photolysis reaction rates currently included within the model. A description of the scattering method employed for this calculation is taken from Hayman (1997) and utilised in Saunders et al. (2003). These values are then attenuated according to the glass type chosen by the user, through the use of a transmission factor ($\psi$).

Indoor photolysis rate values ($\phi$) are available for seven light types, and these are are summed with the attenuated outdoor values to give the total indoor photolysis rate as shown in Eq. (8), where parameters $l$, $m$ and $n$ are optimised as per the discussion in Jenkin et al. (1997).

$$J = ((l\cos(\theta)^m \exp(-n\sec(\theta))\psi) + \phi \tag{8}$$

Values for $\psi$ and $\phi$ are calculated and discussed in Wang et al. (2022). In brief, for the transmission factor, $\psi$, three typical glass types were selected from the analysis in Blocquet et al. (2018), covering a range of transmission values in both the UV (300 - 400 nm) and visible (400 - 800 nm) spectral ranges. The wavelength ranges were split into 10 nm intervals and the percentage of light transmitted through each glass type and for each wavelength interval was defined. The absorption cross-section and quantum yields for each photolysing species, were used to calculate the weighted transmission factor, $\psi$, for each wavelength range For the indoor photolysis rate, $\phi$, the same wavelength ranges, intervals, quantum yields and cross-sections were used, with spherically integrated photon fluxes from Kowal et al. (2017). Seven indoor light types are included in INCHEM-Py, covering a range of transmission spectra. The lights can be set to be on or off by the user for any period, over single or multiple days. Details of the specific lights simulated are found in Kowal et al. (2017); Wang et al. (2022).

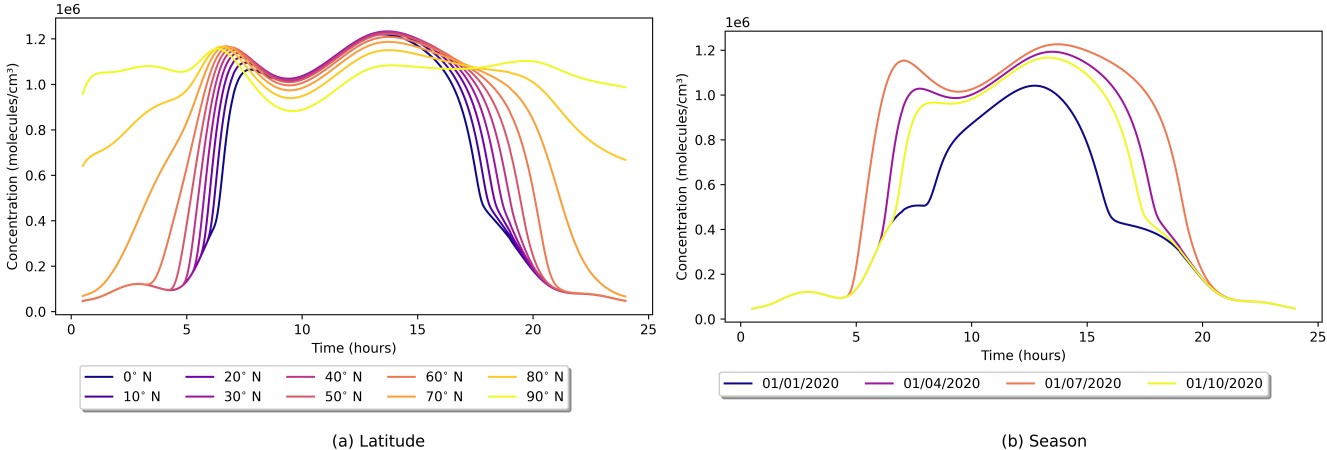

**Figure 2.** Indoor OH concentrations as a function of (a) latitude and (b) time of year. In both cases, only one setting is varied from default, with the latitude variation being on 21/06/2020 and the seasonal variation being at 45° N.

The impact of the different photolysis parameters is shown in Fig. 2 and Fig. 3, where latitude and date settings are varied while all others remain at their default values (given in appendix B). Figure 2(a) shows the effect of latitude variation in the northern hemisphere in summer on indoor OH concentrations. In moving away from the equator, the length of the days increases, but the intensity of sunlight decreases. Therefore, this interplay results in peak indoor OH concentrations at mid

latitudes. At the northern extremes, the Sun does not set, therefore, OH is produced through indoor photolysis reactions for the full 24-hour period but at a reduced rate due to the lower intensity of solar radiation. Figure 2(b) shows the seasonal change of indoor OH in the northern hemisphere. As the outdoor concentrations of species are identical in each simulation the OH concentration difference is solely from photolysis. Reactant concentrations which form OH through photolysis are therefore very similar, resulting in comparable peak concentrations between the simulations.

Figure 3(a) shows the minimal impact of indoor lighting on the concentration of OH indoors, which contrasts with the large effects seen when the glass type is varied in Fig. 3(b). The different glass types impact the production of OH indoors, mainly through the photolysis of HONO and the reaction of $HO_2$ and NO. Glass C lets through the most sunlight and, therefore, allows the most photolysis reactions to occur. At the other end of the scale is the low-emissivity glass with a reflective film that blocks out the majority of wavelengths and is very close to not having any sunlight enter the room at all. Without sunlight there is very

little OH produced by the indoor lighting, which is set by default to come on at 7am and go off at 7pm.

## 2.3 Outdoor air exchange

In INCHEM-Py only indoor species are predicted, and in some cases their concentrations will depend heavily on influx from, and efflux to, outdoors (Kruza et al., 2021). All indoor species are set to decay at a rate of $\lambda_r C_i$ and to increase at a rate of $\lambda_r C_{i,out}$ as in Eq. (1), however only species for which measured, representative outdoor concentrations are available, are as-



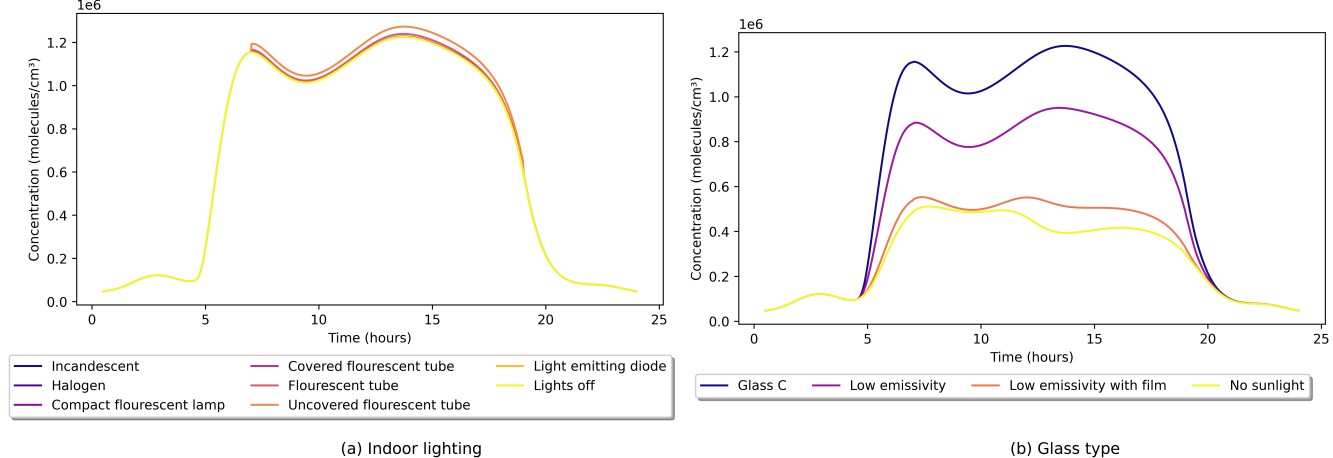

(a) Indoor lighting  (b) Glass type

**Figure 3.** Indoor OH concentrations as a function of (a) indoor light source type and (b) glass type. All other values remained at their default settings.

signed an outdoor concentration in the model. All other species (mostly short-lived intermediates) have outdoor concentrations set to zero.

Concentrations of OH, HO$_2$ and CH$_3$O$_2$ outdoors are photolysis driven and typically show strong diurnal variation outdoors. As in Carslaw (2007), the outdoor concentrations are set to peak at solar noon in the model, at $5 \times 10^6$, $1 \times 10^8$ and $2.5 \times 10^7$ molecules cm$^{-3}$, respectively, in line with measurements taken by Platt et al. (2002) and Emmerson et al. (2005). HONO
shows the opposite trend, as it is photolysed during the day and is set to a minimum of $\approx 20$ ppt at noon, peaking overnight at $\approx 300$ ppt (Alicke, 2003).

NO, NO$_2$ and O$_3$ have four possible diurnal profiles which can be chosen by the user, depending on their specific requirements. Profiles are provided for urban London (UK), suburban London (UK), and urban Bergen (Norway) locations, based on measured data from 2018 in the EEA (2018) air quality database. Details of these sites and their exact identifiers, including
latitude and longitude, are given in the INCHEM-Py user manual. The data from each site were provided as hourly averages and in local time (UTC) by the meteorological station. Using the station longitude, each data point was shifted to solar time and quarter three (Q3) data extracted. Q3 (July, August, September) was chosen because not all data sets had annual data for all species. The daily measurements were then overlaid and an average data point for each hour was used to fit a trigonometric Fourier function for each location and species. The same process was used for outdoor concentrations of PM$_{2.5}$, which can also
be included within the model when the particle module is enabled. The fourth location is Milan (Italy) which is included from Terry et al. (2014) as a particularly polluted two-week period in August 2003. The measurements taken underwent the same procedure of averaging and fitting as the other three locations. The functions that have been input into the model are shown in the user manual. For INCHEM-Py v1.2 the constant outdoor concentrations have been updated, as shown in Table A1. These values have been sourced from published literature and measurement databases as referenced.



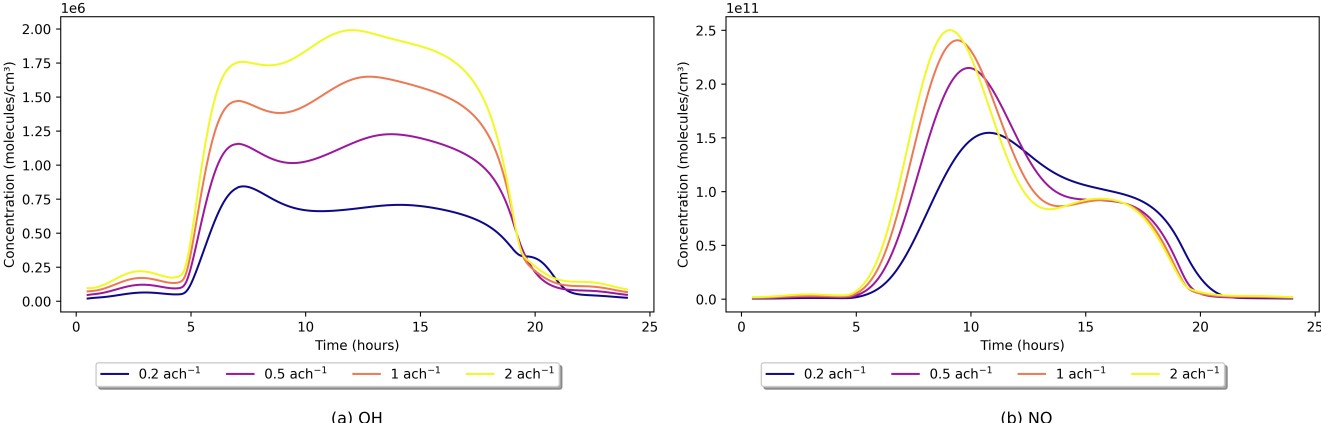

**Figure 4.** Indoor (a) OH and (b) NO concentrations as a function of air exchange rate. All other settings are the default values with the city outdoor concentration profile set to urban Bergen.

The rate of air change (ACR) is a user defined variable that is input in units of $acs^{-1}$ (air changes per second), and will depend on how airtight the space being simulated is. Weschler (2000) gives typical values of between 0.2 and 2 $ach^{-1}$ (air changes per hour) for loosely constructed and tightly constructed residential properties respectively. The default value for INCHEM-Py is $1.38 \times 10^{-4}$ $acs^{-1}$ (0.5 $ach^{-1}$). How the air change rate effects indoor OH concentrations (with all other values set to default) is shown in Fig. 4(a).

In general, increased ACR results in increased OH concentrations. The lifetime of OH is too short to survive ingress through a building and any OH indoors is made indoors through chemical reaction. Through the analysis of the reaction rates, as detailed in Sect. 2.7, OH is mainly produced through $HO_2 + NO \rightarrow OH + NO_2$, and the photolysis reaction $HONO \rightarrow OH + NO$, where $HO_2$ is the hydroperoxy radical that is primarily produced via the oxidation of VOCs. Most indoor NO comes from the photolysis reaction $NO_2 \rightarrow NO + O$ in the absence of indoor sources, and $NO_2$ is mainly produced by the reaction of $NO + O_3 \rightarrow NO_2$. NO, $NO_2$ and $O_3$ are also sourced from outdoors, so as the ACR increases, their concentrations increase, as do the rates of the reactions that produce higher OH as described above.

In the 0.2 $ach^{-1}$ case, there is a small peak in OH concentration at 20:00 which is not seen for the other ACRs. Analysing the reaction rates revealed that as ACR is reduced, indoor $O_3$ concentrations are reduced (as less is available to ingress from outdoors), which decreases the loss of NO through the reaction with $O_3$. This reduced NO consumption results in increased NO concentrations later in the day, as shown in Fig. 4(b), compared to the higher ACR cases. Increased NO concentrations increase the rate of the $NO + HO_2 \rightarrow OH + NO_2$ reaction relative to the other simulations, therefore, causing the peak in OH concentrations later in the day in the low ACR case.



## 2.4 Particles

Gas-to-particle partitioning is included within INCHEM-Py for limonene, $\alpha$-pinene and $\beta$-pinene. Carslaw et al. (2012) provide

a full description of the method used to calculate the particle partitioning parameters, which is based on absorptive partitioning

Pankow (1994), whereby the phase of the species is determined by thermodynamic equilibrium. It is described briefly here as

implemented in the model.

The method relies on a balance between the rate of VOC absorption to ($k_{on}$) and the rate of desorption from ($k_{off}$) particles.

The partitioning coefficient, $K_p$ ($m^3 \mu g^{-1}$) is the gas:particle ratio for a species.

$$K_p = \frac{k_{on}}{k_{off}} \tag{9}$$

This can be calculated using

$$K_p = \frac{7.501RT}{W_{om}10^9 P_l} \tag{10}$$

where $R$ is the ideal gas constant ($J K^{-1} mol^{-1}$), $T$ is the temperature (K), $W_{om}$ is the mean molecular weight of the par-

ticle (g $mol^{-1}$) and $P_l$ is the liquid vapour pressure of the species (Torr). $k_{on}$ is set at a temperature independent value of

$6.2 \times 10^{-3}$ m$^3$ $\mu g^{-1}$ s$^{-1}$, which is based on Jenkin (2004) and Johnson et al. (2006). Rearranging Eq. (9) and substituting Eq.

(10) allows the determination of $k_{off}$ for each species $W_{om}$ is set to 120 g $mol^{-1}$ initially, but is calculated for each subsequent

integration step based on the composition of the formed particles.

Particles can also enter from outdoors with the outdoor particles assumed to be 30 % organic and 70 % inorganic. The organic

fraction from outdoors is then used as a seed on which new indoor particles can form. Each species is tracked individually in

the particle phase, and summed to produce a total number (tsp, molecule cm$^{-3}$) and concentration (tspx, $\mu g$ m$^{-3}$) of suspended

particles.

## 2.5 Temperature

Most chemical reactions in INCHEM-Py have a temperature dependence. Indoor temperatures in most scenarios will have

minimal variation but in some cases might vary throughout a day. The model allows for three methods of setting the indoor

temperature: a constant value, a linear interpolation between given temperatures, or a zero-degree B-Spline interpolation be-

tween given temperatures. For both interpolation methods, a minimum of two temperatures at two distinct times must be given.

INCHEM-Py compares the given times to the length of the simulation and duplicates points before and after the simulated

period, if required. This ensures that there is an interpolated or given temperature value at all time points. Full details of the

methods used are given in the INCHEM-Py user manual. Three different temperature profiles using the three different methods

are shown in Fig. 5, alongside their impact on the OH concentrations. Both interpolated methods used the same times and

temperature inputs.





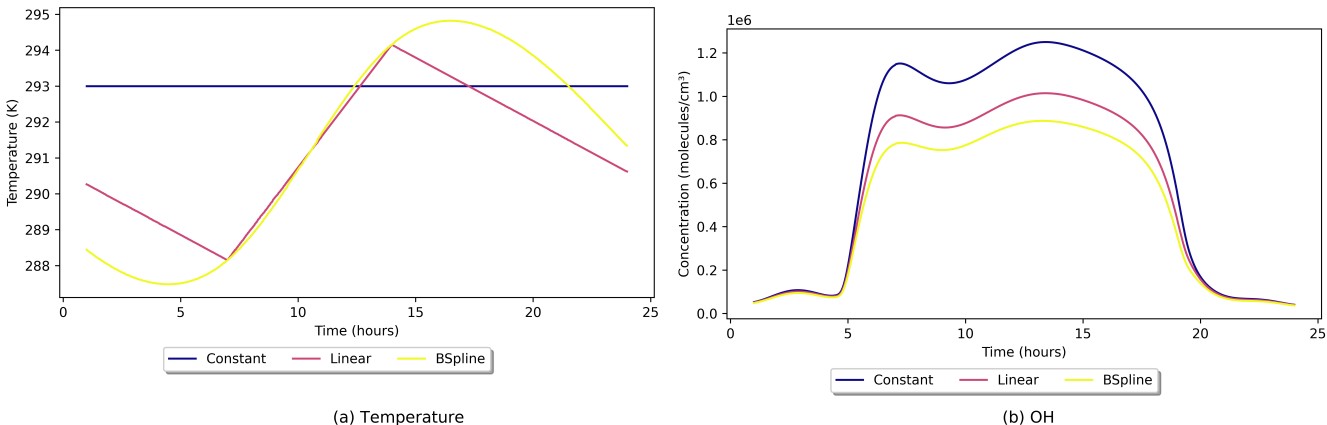

(a) Temperature                                                      (b) OH

**Figure 5.** Results of the three different temperature input methods on (a) temperature and (b) OH concentration. Only the "spline" variable was adjusted from default values, the constant temperature method and value is the default option.

## 2.6 Timed emissions

Term four in Eq. (1) accounts for user-defined timed emissions that can be input to simulate a release of a chemical species over a specific period of time. Only species that are solved by the model can be input, these include chemical and particle species. Outdoor species cannot be input this way, as outdoor concentrations are not solved, but instead are input as time dependent functions or constants which can be adjusted in the outdoor concentrations file. An example of this function is shown in Fig. 6, where an emission of limonene and $\beta$-pinene simulates the use of a cleaning product. Both species were input at a rate of $5 \times 10^8$ molecule cm$^{-3}$ s$^{-1}$ (2 ppt s$^{-1}$) for 10 minutes at 1 pm, while all other settings remained at their default values. Figure 6(a) shows that, the terpene concentrations increase at the same rate and peak at very similar values. The small differences in the subsequent terpene decays, is due to their different reactivities. For example, the reaction rate coefficients at 293K for limonene with O3 ($k_{O_3} = 2.0 \times 10^{-16}$ cm$^3$ molecule$^{-1}$ s$^{-1}$) and OH ($k_{OH} = 1.7 \times 10^{-10}$ cm$^3$ molecule$^{-1}$ s$^{-1}$) are higher than for $\beta$-pinene ($k_{O_3} = 1.8 \times 10^{-17}$ cm$^3$ molecule$^{-1}$ s$^{-1}$, $k_{OH} = 8.1 \times 10^{-11}$ cm$^3$ molecule$^{-1}$ s$^{-1}$) (Jenkin et al., 2018). This explains the lower peak concentration, and the faster decay, for limonene, compared to $\beta$-pinene.

Figure 6(b) shows the secondary production of formaldehyde following the timed emission of limonene and $\beta$-pinene, compared with a background default value with no emissions. Using this method, we can simulate events where multiple chemical species interact, to inform what proportions of an experimentally measured chemical are produced by primary emission or via secondary chemistry.

## 2.7 Reaction rate outputs

When analysing chemical transformations it is useful to know the rate at which a species is being produced or lost, and the relative importance of the individual reactions that are contributing to that rate. INCHEM-Py has an option to output the rates




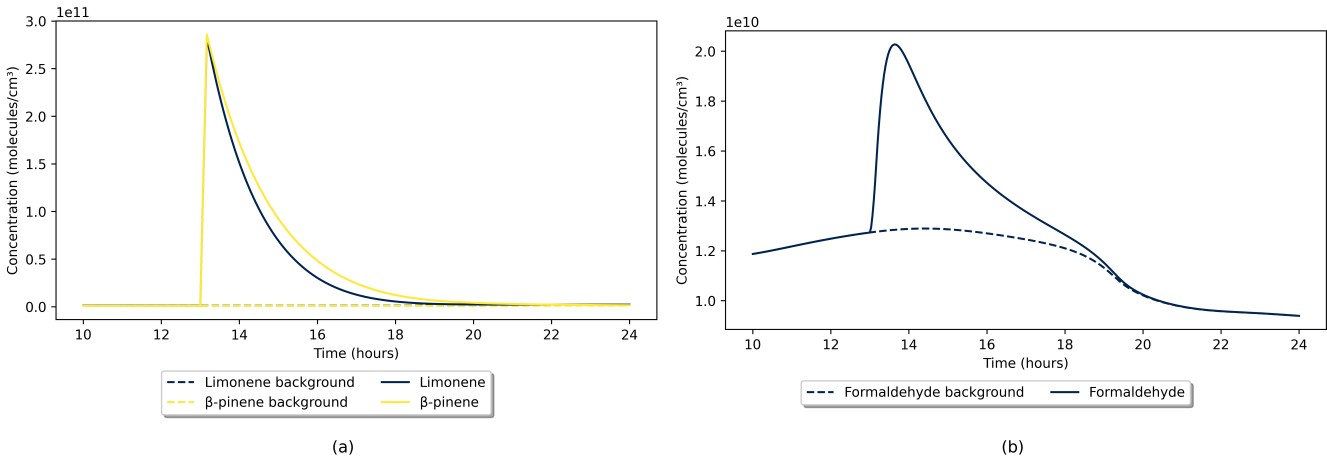

**Figure 6.** (a) Limonene and $\beta$-pinene timed emissions as a simulated cleaning event with (b) formaldehyde secondary emission compared to background default values.

of all reactions at all time points. Using the rate coefficients, and the calculated species concentrations, the reaction rates of individual reactions can be calculated at any time point in the simulation using Eq. (3). This function was developed for, and used by, Lakey et al. (2021) to identify important reactions and develop a reduced chemical mechanism for use in an indoor 3D fluid model.

In the case of the cleaning event shown in Fig. (6), the reaction rates from INCHEM-Py at each time step can be used to track the pathways linking the primary pollutants (limonene and $\beta$-pinene) to the secondary production of formaldehyde. This is visualised in Fig. 7, where a snapshot of the reaction pathways at the peak of the pollutant concentrations is shown. This method was used to analyse the contribution of surface cleaner formulations to indoor pollutants in Carslaw and Shaw (2022).

## 2.8   Surface Deposition

Surface deposition of gas-phase species is an important aspect of indoor air chemistry, and is key to include in indoor air models. Chemical species can be emitted from surfaces, either as primary emissions, or as secondary pollutants formed from gas-phase transformations which occur at surface level. $O_3$ can deposit onto a range of surfaces and induce oxidation, releasing secondary pollutants as surface emissions (Gall et al., 2013; Cros et al., 2012). The rate of deposition to surfaces is surface-specific, and is determined by mass transportation to the surface of a pollutant, and the uptake potential of the pollutant onto

that specific surface(Reiss et al., 1994). The deposition rate of an oxidant onto a surface is also influenced by the air change rate, the bulk indoor concentration of the oxidant, and the surface-to-volume ratios in the indoor environment (Coleman et al., 2008).





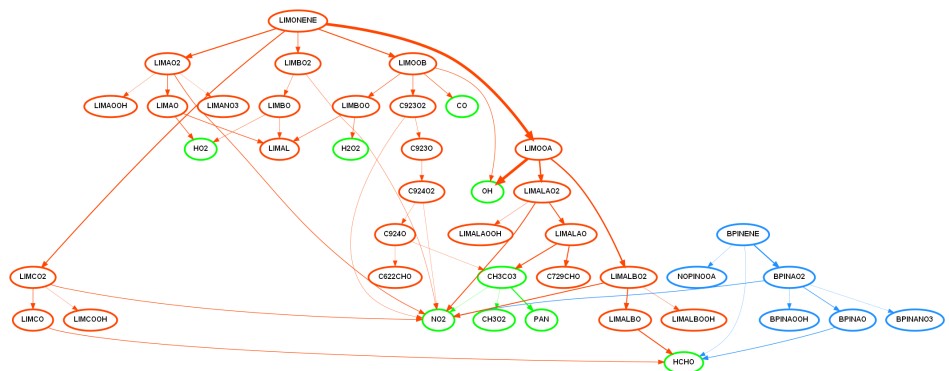

**Figure 7.** A flow chart showing the degradation pathways of limonene (orange) and $\beta$-pinene (blue) to final secondary products of interest (green). The fastest 95% of reactions occurring at the time point where their concentrations are highest in Fig. 6 are shown. The thickness of the line is proportional to the absolute reaction rate, where the thicker the line, the faster the reaction. Nodes show species created, and all species are labelled in MCM format.

INCHEM-Py v1.1 simulates the irreversible deposition of 3371 indoor gas-phase species. However, it does not incorporate the secondary pollutants emitted from the surface. It also does not consider different surface materials, and only calculates
surface deposition based on estimated deposition velocities and a total surface to volume ratio (Carslaw, 2007).

A new surface deposition mechanism onto multiple surfaces has been developed for INCHEM-Py v1.2, based on the work of Kruza et al. (2017) and considering the rates of deposition and secondary pollutant emissions following individual $O_3$ and $H_2O_2$ deposition (Carter et al., 2023). The surface removal rate of $O_3$ and $H_2O_2$ in the model is determined by Eq. (11) and Eq. (12), respectively:

$$F_{O_3} = \nu_{d_{O_3}} \frac{A}{V} \tag{11}$$

$$F_{H_2O_2} = \nu_{d_{H_2O_2}} \frac{A}{V} \tag{12}$$

where $F_{O_3}$ and $F_{H_2O_2}$ represent the deposition rates (s$^{-1}$) of $O_3$ and $H_2O_2$ onto a surface, respectively. $\nu_d$ represents the surface deposition velocity of an oxidant in cm s$^{-1}$, $A$ is the surface area of an indoor surface or material in cm$^2$ and $V$ is the
total volume of the indoor environment in cm$^3$.

The emission rate calculations of secondary pollutants emitted as a result of oxidant surface deposition has been adapted from studies conducted by Morrison and Nazaroff (2002) and Kruza et al. (2017). The emission rates can be determined by solving Eq. (13) and Eq. (14):

$$E_i = \frac{A\nu_{d_{O_3}} Y_i C_{O_3}}{V} \tag{13}$$


$$E_i = \frac{A\nu_{d_{H_2O_2}} Y_i C_{H_2O_2}}{V} \tag{14}$$





where $E_i$ represents the emission of secondary pollutants from a surface (molecule cm$^{-3}$ s$^{-1}$). $Y$ is the production yield of gas-phase species following deposition (dimensionless), and $C_{O_3}$ and $C_{H_2O_2}$ represent the bulk concentrations of indoor O$_3$ and H$_2$O$_2$, respectively (molecule cm$^{-3}$).

Surfaces included in the O$_3$ and H$_2$O$_2$ surface deposition mechanisms for INCHEM-Py v1.2 are soft fabric, painted, human skin, wood, metal, concrete, paper, plastic and glass. However, there are no data for H$_2$O$_2$ onto plastic, glass and skin surfaces. The mechanisms were constructed using surface specific deposition velocities of O$_3$ and H$_2$O$_2$ and respective production yields from a range of experimental literature (Sabersky et al., 1973; Lin and Hsu, 2015; Klenø et al., 2001; Grøntoft, 2002; Abbass et al., 2017; Gall et al., 2013; Tamás et al., 2006; Cros et al., 2012; Coleman et al., 2008; Ye et al., 2020; Lamble et al., 2011;

Rim et al., 2016; Poppendieck et al., 2007; Wang and Morrison, 2010, 2006; Nicolas et al., 2007; Morrison and Nazaroff, 2000; Fadeyi et al., 2013; Yao et al., 2020; Di et al., 2017; Rai et al., 2014; Fischer et al., 2013; Wisthaler and Weschler, 2010; Schripp et al., 2012; Mueller et al., 1973; Cox and Penkett, 1972; Grøntoft and Raychaudhuri, 2004; Simmons and Colbeck, 1990; Cano-Ruiz et al., 1993; Poppendieck et al., 2021). These mechanisms ensure secondary species, primarily aldehydes, are emitted from specific surfaces as a result of oxidant deposition. A deposition velocity taken from Carslaw (2007) has been

added for plastic, glass and skin surfaces to account for gas-phase deposition of H$_2$O$_2$. Deposition of H$_2$O$_2$ onto these three surfaces do not produce any aldehyde emissions in our model.

    A detailed description of the oxidant surface deposition mechanisms is found in a corresponding study conducted by Carter et al. (2023). Using these methods, further deposition mechanisms can be developed for other species and surfaces in the future, as relevant experimental data become available.

Since the Carter et al. (2023) study, the deposition module for INCHEM-Py has been updated to include O$_3$ deposition onto linoleum surfaces, using data from Kruza et al. (2017). The deposition velocity of O$_3$ onto plastic surfaces has also been updated as a result of an ongoing review of the literature. The deposition velocities of O$_3$ onto linoleum and plastic surfaces are now 0.0070 and 0.1225 cm s$^{-1}$ respectively (Coleman et al., 2008; Kruza et al., 2017; Klenø et al., 2001; Poppendieck et al., 2007; Nicolas et al., 2007; Wang and Morrison, 2006).

**2.9   Direct emissions**

Breath emission values from humans are optionally included in the model, according to occupancy status (Kruza and Carslaw, 2019; Weschler et al., 2007). The number of adults and children can be specified in the settings file and a calculated emission in molecule cm$^{-3}$ is used, as shown in Table 1. These emissions are constant for the duration of the simulation.

    **3   Model output and evaluation**

In this section, the INCHEM-Py model is used to simulate previously published results from both experimental and modelling efforts. Settings and outputs from all of the model runs are linked in the data availability section of this paper.



**Table 1.** Rates of breath and skin emissions for adults and children

| Emitted species | Adult rate (molecule $cm^{-3}$) | Child rate (molecules $cm^{-3}$) |
| --- | --- | --- |
| Acetone | $2.534 \times 10^7$ | $4.781 \times 10^6$ |
| Ethanol | $1.988 \times 10^7$ | $3.009 \times 10^6$ |
| Methanol | $8.512 \times 10^6$ | $3.108 \times 10^6$ |
| Isopropanol | $3.862 \times 10^6$ | $6.593 \times 10^5$ |
| Isoprene | $5.412 \times 10^6$ | $5.953 \times 10^5$ |

## 3.1 Skin oil emissions

In Kruza and Carslaw (2019), the experimental results showing 4-oxopentanal (4-OPA) production from human skin from Weschler et al. (2007) and Wisthaler and Weschler (2010) were simulated using the precurser to INCHEM-Py, the INDCM.
The experiment involved two adults (without showering or having used personal care products) being placed in a 28.5 $m^3$ chamber with a high level of $O_3$. Concentrations of 4-OPA produced from surface reactions on human skin and clothing were then measured. The original work used units of ppb so outputs from INCHEM-Py were duly converted for this comparison.

Two simulations were run in INCHEM-Py. The first was optimised to achieve a constant background $O_3$ concentration of 32 ppb to match that measured by Wisthaler and Weschler (2010). As the experimental chamber was located inside a laboratory,
the diurnal outdoor concentrations were disabled in the model, and an emission of $O_3$ was created. The experimental chamber, a former aircraft cabin, had painted plastic walls and a carpeted floor giving surface area to volume ratios of 0.004 $cm^{-1}$ for soft furnishings and 0.0158 $cm^{-1}$ for painted walls. An emission of $9 \times 10^8$ molecule $cm^{-3}$ $s^{-1}$ (0.037 ppb $s^{-1}$) of $O_3$ was needed to maintain a 31.53 ppb $O_3$ concentration with an air change rate of 1 $h^{-1}$. To match the experimental conditions, the simulated chamber was set to have constant indoor lighting from incandescent bulbs and no sunlight, the temperature was set
to 296.15 K, and the relative humidity was 23 %.

With the background simulation optimised, a second simulation was run using the background simulation outputs as initial concentrations. From 10 am to 2 pm, two humans with a combined surface to volume ratio of 0.0014 $cm^{-1}$ (assuming a skin area of 2 $m^2$ per person) and associated breath emissions (according to Tab. 1), were simulated to be present in the room. The decrease in $O_3$ and production of 4-OPA from the skin surface chemistry from experiment and the two models is shown
in Fig. 8. The INCHEM-Py model output shows good agreement with the experimental results with the $O_3$ decreasing at the same rate but not for quite as long. When comparing the results from INCHEM-Py and the INDCM, the $O_3$ concentrations share the same shape, but due to the improved surface mechanism, a balance between $O_3$ deposition and 4-OPA yield results in much closer alignment to the experimental data. The 4-OPA measurements increase more slowly than the INCHEM-Py concentration, which is probably a result of the lack of spatial representation in the model. 8.
With no diurnal variation in the INCHEM-Py simulation, the ratio of $O_3$ to 4-OPA will eventually reach equilibrium, determined by the surface to volume ratio, the surface deposition velocity, and the 4-OPA yield from $O_3$-surface interactions





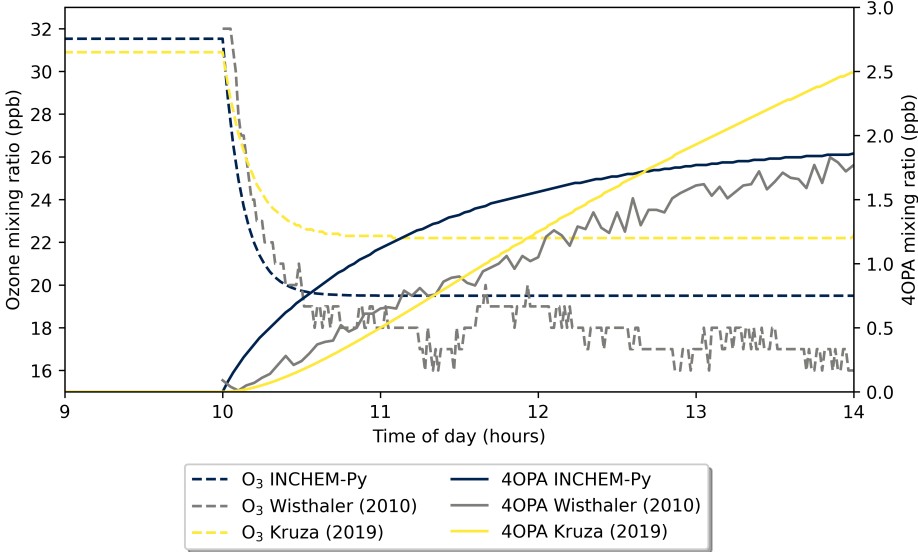

**Figure 8.** Comparison of experimental (Wisthaler, grey lines) and simulated results (INCHEM-Py, black lines, and Kruza, yellow lines) of 4-OPA production from skin and clothing due to $O_3$ surface chemistry.

as in Eq. (13). Therefore, using INCHEM-Py and repeated experimental measurements of different surfaces, the values for these constants can be validated using the ratio of $O_3$ to product at equilibrium. Given experimental uncertainties, the values currently present within the model give a good estimation of 4-OPA production with minimal changes to the INCHEM-Py
settings file.

## 3.2 $H_2O_2$ emission and fate

Zhou et al. (2020) studied the impact of non-bleach cleaning events in a combined chamber and modelling study. In this study, the INDCM model was used to explore secondary pollutant formation, and the impact of different lighting levels on their concentrations. Again, the original work was completed in ppb and so outputs from INCHEM-Py have been converted for the
comparison.

The Indoor Environmental Quality chamber at the Building Energy and Environmental Systems Laboratory at Syracuse University had a mock residential room built within it. This 29.1 $m^3$ room had a wooden frame, painted walls and a vinyl floor with no furniture present. Within this chamber a non-bleach $H_2O_2$ cleaning spray was used on a 0.75 $m^2$ area (12 squirts, 15 mL total) and then wiped dry over a period of 1-2 minutes. The surface to volume ratio of the experiment simulated here is
2 $m^{-1}$ and the floor is assumed to be 1/6 of the total surface area.

The air change rate was constant at 0.51±0.004 $h^{-1}$ and was sourced from outdoors, the temperature was controlled at 25.7±0.9 °C and relative humidity averaged at 25.8±9.5 % during the experiments. The indoor lighting was from uncovered fluorescent tubes and a solar illuminator was used to provide outdoor lighting through the window of the experimental room.





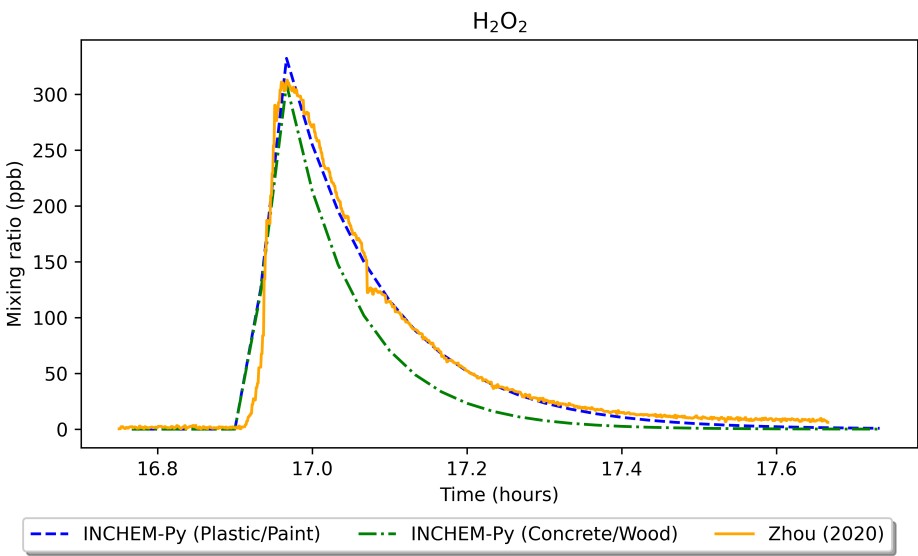

**Figure 9.** Comparison of experimental results of a $H_2O_2$ cleaning event from Zhou et al. (2020) and simulated results using INCHEM-Py.

The solar illuminator removed the diurnal variation of the photolysis but not of the outdoor concentrations. In the experiment,
the solar flux was measured using a spectrometer, and photolysis coefficients of key species ($H_2O_2$, $NO_2$, HONO, $NO_3$, $O_3$ and formaldehyde) were calculated and input into the INDCM. In the INCHEM-Py simulations, to account for the constant solar photolysis, the local hour angle (LHA in inchem_main.py, described in section 2.2) was set to 0 to give constant noon photolysis values. A timed emission rate of $5.5 \times 10^{10}$ molecule $cm^{-3}$ $s^{-1}$ (2.25 ppb $s^{-1}$) for 3 minutes was used to simulate the cleaning spray, based on the measured values.

Figure 9 shows the $H_2O_2$ mixing ratio of the experiment from Zhou et al. (2020) and two INCHEM-Py model runs - one simulating the experimental conditions of a plastic floor and painted walls (plastic/paint), and a second where the floor is changed to concrete and the walls to wood (concrete/wood). The plastic/paint scenario shows good agreement between the simulated and measured $H_2O_2$ mixing ratios. At peak $H_2O_2$ concentration, the main loss of $H_2O_2$ was to the painted walls ($1.54 \times 10^{10}$ molecule $cm^{-3}$ $s^{-1}$), followed by loss to the plastic floor and loss to outdoors ($1.43 \times 10^9$ and
$1.16 \times 10^9$ molecule $cm^{-3}$ $s^{-1}$, respectively). Compared to the plastic/paint scenario, the predicted $H_2O_2$ mixing ratio peaks at a lower value, and drops more quickly after the cleaning, in the concrete/wood scenario. This is due to an larger loss rate to the wooden walls ($1.64 \times 10^{10}$ molecule $cm^{-3}$ $s^{-1}$) and concrete floor ($5.96 \times 10^9$ molecule $cm^{-3}$ $s^{-1}$) at peak $H_2O_2$ concentration.

Following the cleaning event, formaldehyde is produced as a secondary product. Using INCHEM-Py, the maximum increase in formaldehyde as a result of secondary chemistry in the plastic/paint scenario was predicted to be 0.72 ppb. In the experi-
ment (Zhou et al. (2020)), the secondary emission peak was masked by the background formaldehyde concentration of over 50 ppb, therefore, this shows the importance of modelling for delineating between primary and secondary emissions. In the concrete/wood scenario, the increase in formaldehyde as a result of secondary chemistry was much higher, at 3.05 ppb. This





is due to higher $H_2O_2$ deposition velocities, and higher resulting surface emissions from concrete/wood surfaces, compared to the plastic/paint combination (Carter et al., 2023).

**4   Conclusions**

INCHEM-Py (the INdoor CHEMical model in Python) has been presented as a tool for the analysis and interrogation of the atmospheric chemistry of the indoor environment. We have presented the main modules developed for v1.2 within this paper, and have given a further detailed description in the user manual submitted alongside. INCHEM-Py has been developed as an open and accessible piece of software, has no hidden or proprietary code, and requires minimal previous coding or modelling

experience to install or run. It utilises core Python libraries, including Numpy (Harris et al., 2020) and Scipy (Virtanen et al., 2020), keeping installation of additional libraries to a minimum, and capitalising on the maintenance of the Python ecosystem.

INCHEM-Py has been validated against experimental measurements and has shown improved accuracy in comparison with the INDCM, from which it was refactored. INCHEM-Py has been developed with a focus on predicting secondary chemistry that is not feasible, or in some cases possible, to measure. Outputs of species concentrations with time are given alongside key

model parameters such as surface deposition rates, seasonal photolysis rates and diurnal outdoor concentrations. Interrogating these outputs allows for a detailed understanding of the atmospheric chemical processes that occur indoors, including temporally resolved reaction rates which can be used to identify the important pathways that should be included in models with reduced chemical mechanisms (Lakey et al., 2021).

The utility of INCHEM-Py is further demonstrated by the publications that have used the model since the Shaw and Carslaw

(2021) release of v1.1. Published articles include Lakey et al. (2021); Wang et al. (2022); Carter et al. (2023); Beel et al. (2023), with several additional articles in preparation that discuss emissions from cooking and cleaning, UV induced emissions from plastics, and future indoor air pollution scenarios. Each case exhibits the versatility of INCHEM-Py and the ability of users to simulate custom experimental scenarios to expand our understanding of indoor air chemistry.

Future developments of INCHEM-Py will likely include an expansion of the gas-to-particle partitioning module to include

more source species and prediction of particle size distributions. Chemical mechanisms and analysis tools will continue to be added as they are developed, including a Python package for the analysis of INCHEM-Py outputs. In the long term a multi-box approach may be taken to simulate adjoining spaces, or a hybrid approach may be used to add spatial dimensions.



**Appendix A: Outdoor constant concentrations and source of measurements on which they are based**

| Species | v1.1 value (molecule/cm$^3$) | v1.2 value (molecules/cm$^3$) | Ref. |
|---|---|---|---|
| Formaldehyde | $9.13 \times 10^{10}$ | $6.017 \times 10^{10}$ | Uchiyama et al. (2015) |
| Acetaldehyde | $7.15 \times 10^{10}$ | $3.896 \times 10^{10}$ | Uchiyama et al. (2015) |
| Propanal | $2.02 \times 10^{10}$ | $9.332 \times 10^{09}$ | Uchiyama et al. (2015) |
| 3-Methylbutanal | - | $1.049 \times 10^{09}$ | Uchiyama et al. (2015) |
| Acrolein | $4.94 \times 10^{10}$ | $2.685 \times 10^{09}$ | Uchiyama et al. (2015) |
| Methacrolein | - | $2.792 \times 10^{09}$ | Baudic et al. (2016) |
| Crotonaldehyde | - | $1.718 \times 10^{09}$ | Uchiyama et al. (2015) |
| Pentanal | - | $2.447 \times 10^{09}$ | Uchiyama et al. (2015) |
| Hexanal | $9.25 \times 10^{09}$ | $2.706 \times 10^{09}$ | Uchiyama et al. (2015) |
| Heptanal | $3.75 \times 10^{09}$ | $1.846 \times 10^{09}$ | Uchiyama et al. (2015) |
| Octanal | $7.25 \times 10^{09}$ | $2.349 \times 10^{09}$ | Uchiyama et al. (2015) |
| Nonanal | $2.5 \times 10^{10}$ | $1.482 \times 10^{10}$ | Uchiyama et al. (2015) |
| Decanal | $2.75 \times 10^{09}$ | $4.047 \times 10^{09}$ | Uchiyama et al. (2015) |
| 2-Nonenal | - | $1.288 \times 10^{09}$ | Uchiyama et al. (2015) |
| Acetone | $1.3 \times 10^{10}$ | $4.977 \times 10^{10}$ | Uchiyama et al. (2015) |
| 2-Butanone (MEK) | $2.41 \times 10^{09}$ | $5.429 \times 10^{09}$ | Uchiyama et al. (2015) |
| 3-Buten-2-one (MVK) | $1.78 \times 10^{10}$ | $2.792 \times 10^{09}$ | Baudic et al. (2016) |
| Cyclohexanone | $9 \times 10^{08}$ | $1.706 \times 10^{10}$ | Lü et al. (2006) |
| Benzaldehyde | $6.13 \times 10^{10}$ | $1.419 \times 10^{09}$ | Uchiyama et al. (2015) |
| o-Tolualdehyde | - | $1.253 \times 10^{09}$ | Uchiyama et al. (2015) |
| m-Tolualdehyde | - | $2.005 \times 10^{09}$ | Uchiyama et al. (2015) |
| p-Tolualdehyde | - | $2.005 \times 10^{09}$ | Uchiyama et al. (2015) |
| 2,5-Dimethylbenzaldehyde | - | $7.854 \times 10^{09}$ | Uchiyama et al. (2015) |
| Benzene | $5.9 \times 10^{09}$ | $9.637 \times 10^{09}$ | Uchiyama et al. (2015) |
| Toluene | $2 \times 10^{10}$ | $4.085 \times 10^{10}$ | Uchiyama et al. (2015) |
| p-Xylene | $6.5 \times 10^{09}$ | $6.098 \times 10^{09}$ | Uchiyama et al. (2015) |
| m-Xylene | $6.5 \times 10^{09}$ | $6.098 \times 10^{09}$ | Uchiyama et al. (2015) |
| o-Xylene | $1.3 \times 10^{10}$ | $4.254 \times 10^{09}$ | Uchiyama et al. (2015) |
| Ethylbenzene | $3.4 \times 10^{09}$ | $8.792 \times 10^{09}$ | Uchiyama et al. (2015) |
| Propylbenzene | - | $4.008 \times 10^{09}$ | Mentese and Bas (2020) |

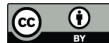



| Species | v1.1 value (molecule/cm$^3$) | v1.2 value (molecules/cm$^3$) | Ref. |
|---|---|---|---|
| 2-Ethyltoluene | - | $2.505\times10^{08}$ | Bari and Kindzierski (2018) |
| 3-Ethyltoluene | - | $6.013\times10^{08}$ | Bari and Kindzierski (2018) |
| 4-Ethyltoluene | - | $3.006\times10^{08}$ | Bari and Kindzierski (2018) |
| 1,3,5-Trimethylbenzene | $2.2\times10^{10}$ | $1.754\times10^{09}$ | Uchiyama et al. (2015) |
| 1,2,4-Trimethylbenzene | $2.23\times10^{09}$ | $5.511\times10^{09}$ | Uchiyama et al. (2015) |
| 1,2,3-Trimethylbenzene | $8.0\times10^{09}$ | $1.253\times10^{09}$ | Uchiyama et al. (2015) |
| p-Dichlorobenzene | - | $1.229\times10^{10}$ | Uchiyama et al. (2015) |
| Styrene | $5.67\times10^{09}$ | $2.313\times10^{09}$ | Mentese and Bas (2020) |
| Cumene | $1.56\times10^{08}$ | $3.006\times10^{09}$ | Mentese and Bas (2020) |
| Phenol | $5\times10^{10}$ | $1.747\times10^{10}$ | Sturaro et al. (2010) |
| Ethane | $2.08\times10^{10}$ | $9.133\times10^{10}$ | Baudic et al. (2016) |
| Propane | $1.25\times10^{10}$ | $3.797\times10^{10}$ | Baudic et al. (2016) |
| Butane | $3.33\times10^{10}$ | $3.471\times10^{10}$ | Baudic et al. (2016) |
| Isobutane | - | $2.321\times10^{10}$ | Baudic et al. (2016) |
| 2,2-Dimethylbutane | - | $2.027\times10^{09}$ | Bari et al. (2016) |
| 2,3-Dimethylbutane | - | $2.586\times10^{09}$ | Bari et al. (2016) |
| Pentane | $9.13\times10^{09}$ | $8.681\times10^{09}$ | Baudic et al. (2016) |
| 2-Methylpentane | - | $3.844\times10^{09}$ | Bari et al. (2016) |
| 3-Methylpentane | - | $2.446\times10^{09}$ | Bari et al. (2016) |
| Isopentane | - | $1.469\times10^{10}$ | Bari and Kindzierski (2018) |
| Hexane | $9.15\times10^{09}$ | $1.118\times10^{10}$ | Uchiyama et al. (2015) |
| 2-Methylhexane | - | $2.464\times10^{09}$ | Bari et al. (2016) |
| 3-Methylhexane | - | $3.125\times10^{09}$ | Bari et al. (2016) |
| Heptane | $2.58\times10^{09}$ | $6.010\times10^{08}$ | Uchiyama et al. (2015) |
| Octane | $7.5\times10^{09}$ | $5.272\times10^{08}$ | Uchiyama et al. (2015) |
| Nonane | $1\times10^{10}$ | $3.052\times10^{09}$ | Uchiyama et al. (2015) |
| Decane | $1.96\times10^{09}$ | $9.946\times10^{09}$ | Uchiyama et al. (2015) |
| Undecane | $1.95\times10^{09}$ | $1.445\times10^{10}$ | Uchiyama et al. (2015) |
| Dodecane | $5.22\times10^{08}$ | $1.061\times10^{09}$ | Mentese and Bas (2020) |
| Cyclohexane | $1.19\times10^{09}$ | $6.440\times10^{08}$ | Bari and Kindzierski (2018) |
| Ethene | $1.25\times10^{10}$ | $3.327\times10^{10}$ | Baudic et al. (2016) |
| | | | *continued on next page* |






| Species | v1.1 value (molecule/cm$^3$) | v1.2 value (molecules/cm$^3$) | Ref. |
| --- | --- | --- | --- |
| Propene | $4.3\times10^{09}$ | $9.159\times10^{09}$ | Baudic et al. (2016) |
| 1-Butene | - | $3.971\times10^{09}$ | Bari et al. (2016) |
| cis-2-Butene | $3.5\times10^{09}$ | $4.293\times10^{08}$ | Bari and Kindzierski (2018) |
| trans-2-Butene | $4\times10^{09}$ | $5.367\times10^{08}$ | Bari and Kindzierski (2018) |
| 2-Methyl-1-butene | - | $5.152\times10^{08}$ | Bari and Kindzierski (2018) |
| 2-Methyl-2-butene | $7\times10^{09}$ | $4.293\times10^{08}$ | Bari and Kindzierski (2018) |
| Isoprene | $1.0\times10^{10}$ | $2.299\times10^{09}$ | Baudic et al. (2016) |
| 1,3-Butadiene | $2.5\times10^{09}$ | $5.567\times10^{08}$ | Bari and Kindzierski (2018) |
| trans-2-Pentene | - | $4.293\times10^{08}$ | Bari and Kindzierski (2018) |
| cis-2-Pentene | - | $2.576\times10^{08}$ | Bari and Kindzierski (2018) |
| Ethyne | - | $1.573\times10^{10}$ | Baudic et al. (2016) |
| Methanol | $1.3\times10^{11}$ | $1.107\times10^{11}$ | Baudic et al. (2016) |
| Ethanol | $1.2\times10^{12}$ | $1.613\times10^{11}$ | Gallego et al. (2016) |
| Isopropanol | $2.0\times10^{10}$ | $9.239\times10^{10}$ | Gallego et al. (2016) |
| 1-Propanol | $1.2\times10^{09}$ | $1.243\times10^{10}$ | Gallego et al. (2016) |
| 1-Butanol | $1.3\times10^{10}$ | $2.519\times10^{10}$ | Gallego et al. (2016) |
| 1-Pentanol | - | $5.658\times10^{07}$ | Hellén et al. (2018) |
| 1-Hexanol | - | $3.014\times10^{07}$ | Hellén et al. (2018) |
| 2-Butoxyethanol | $5.19\times10^{09}$ | $2.507\times10^{10}$ | Gallego et al. (2016) |
| Linalool | - | $1.292\times10^{07}$ | Hellén et al. (2018) |
| Chloroform | $2.93\times10^{08}$ | $7.567\times10^{08}$ | Uchiyama et al. (2015) |
| Methylchloroform | $8.33\times10^{10}$ | $7.674\times10^{09}$ | Brickus et al. (1998) |
| Dichloromethane | $1\times10^{09}$ | $2.340\times10^{09}$ | Bari and Kindzierski (2018) |
| Trichloroethylene | $3\times10^{07}$ | $9.075\times10^{09}$ | Gallego et al. (2016) |
| Tetrachloroethylene | $2\times10^{08}$ | $5.084\times10^{08}$ | Bari and Kindzierski (2018) |
| 1,2-Dichloroethane | - | $4.260\times10^{08}$ | Bari and Kindzierski (2018) |
| Chloromethane | - | $1.396\times10^{10}$ | Bari and Kindzierski (2018) |
| Hydrogen Chloride | - | $3.716\times10^{10}$ | Uchiyama et al. (2015) |
| Ethyl Acetate | - | $2.392\times10^{09}$ | Uchiyama et al. (2015) |
| Butyl Acetate | - | $1.296\times10^{09}$ | Uchiyama et al. (2015) |
| $\alpha$-Pinene | $1.45\times10^{09}$ | $3.094\times10^{09}$ | Uchiyama et al. (2015) |



| Species | v1.1 value (molecule/cm$^3$) | v1.2 value (molecules/cm$^3$) | Ref. |
|---|---|---|---|
| $\beta$-Pinene | $2.5 \times 10^{07}$ | $1.238 \times 10^{09}$ | Gallego et al. (2016) |
| Limonene | $9 \times 10^{08}$ | $2.431 \times 10^{09}$ | Uchiyama et al. (2015) |
| $\Delta$3-Carene | $9 \times 10^{08}$ | $2.718 \times 10^{09}$ | Hakola et al. (2009) |
| Camphene | $5 \times 10^{08}$ | $3.978 \times 10^{08}$ | Hakola et al. (2009) |
| Formic Acid | - | $1.832 \times 10^{11}$ | Uchiyama et al. (2015) |
| Acetic Acid | - | $3.861 \times 10^{11}$ | Uchiyama et al. (2015) |
| Propanoic Acid | - | $1.873 \times 10^{09}$ | Hellén et al. (2018) |
| Butanoic Acid | - | $1.381 \times 10^{09}$ | Hellén et al. (2018) |
| Pentanoic Acid | - | $7.534 \times 10^{08}$ | Hellén et al. (2018) |
| Heptanoic Acid | - | $9.932 \times 10^{07}$ | Hellén et al. (2018) |
| Hydrogen Peroxide | $5 \times 10^{10}$ | $3.13 \times 10^{10}$ | He et al. (2010) |
| $\beta$-Caryophyllene | $2.5 \times 10^{07}$ | $9.348 \times 10^{07}$ | Hellén et al. (2018) |
| Methane (CH$_4$) | $4.63 \times 10^{13}$ | $4.652 \times 10^{13}$ | Dlugokencky (2022) |
| Carbon Monoxide (CO) | $2.5 \times 10^{12}$ | $6.642 \times 10^{12}$ | Naghizadeh et al. (2019) |
| Sulfur Dioxide (SO$_2$) | - | $1.715 \times 10^{10}$ | EEA (2018) |
| Nitrogen Oxide (NO) | $2.59 \times 10^{10}$ | Average | EEA (2018) |
| Nitrogen Dioxide (NO$_2$) | $9.52 \times 10^{10}$ | Average | EEA (2018) |
| Ozone (O$_3$) | $7.68 \times 10^{11}$ | Average | EEA (2018) |
| Nitric Acid (HNO$_3$) | $5 \times 10^{10}$ | $9.557 \times 10^{09}$ | Vichi et al. (2016) |
| Nitrous Acid (HONO) | $1.6 \times 10^{09}$ | $1.588 \times 10^{10}$ | Vichi et al. (2016) |
| Hydroxyl Radical (OH) | $1 \times 10^{06}$ | $1.09 \times 10^{06}$ | Li et al. (2018) |
| Peroxyacetyl Nitrates (PAN) | $1.51 \times 10^{10}$ | $5.449 \times 10^{10}$ | Liu et al. (2018) |
| Total Suspended Particles (TSP) | $1.4 \times 10^{11}$ | Average | EEA (2018) |

**Table A1.** Outdoor constant concentrations used in INCHEM-Py including the references used to obtain the v1.2 values. Values given as "Average" are averages of the diurnal profiles used in the model.



**400 Appendix B: Default settings**

```
filename = 'mcm_v331.fac'

particles = True

INCHEM_additional = True

custom = False

spline = 293.
temperatures = [[25200,288.15],[50400,294.15]]

rel_humidity = 50.
M = 2.51e+19

const_dict = {
    'O2':0.2095*M,
    'N2':0.7809*M,
    'H2':550e-9*M,
    'saero':1.3e-2}
ACRate =    {0        :   0.5/3600,
            3600 * 24:   1/3600,
            3600 * 48:   2/3600}
diurnal = True
city = "Bergen_urban"

date = "21-06-2020"
lat = 45
light_type="Incand"
light_on_times=[[7,19],[31,43],[55,67],[79,91]]
glass="glass_C"

AV = 0.02
```




```
surfaces_AV = {
          'AVSOFT' : 0.0035,
          'AVPAINT' : 0.0114,
          'AVWOOD' : 0.0061,
          'AVMETAL' : 0.0025,
440       'AVCONCRETE' : 0.0001,
          'AVPAPER' : 0.0006,
          'AVLINO' : 0.0000,
          'AVPLASTIC' : 0.0048,
          'AVGLASS' : 0.0009,
445       'AVHUMAN' : 0.0017}

      H2O2_dep = True
      O3_dep = True

adults = 0
      children = 0

      initials_from_run = False

initial_conditions_gas = 'initial_concentrations.txt'

      timed_emissions = False

      timed_inputs = {"LIMONENE":[[46800,47400,5e8],[107600,108000,5e8]],
460               "BPINENE":[[46800,47400,5e8]]}

      dt = 120
      t0 = 0
      seconds_to_integrate = 86400

      custom_name = "Bergen_urban"

      reactions_output = True
```

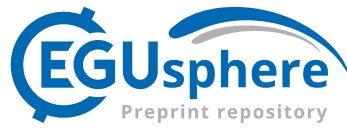

```
output_graph = True
      output_species = ['LIMONENE','APINENE']
```



*Code and data availability.* INCHEM-Py, including the user manual, is available on github at https://github.com/DrDaveShaw/INCHEM-Py with an archived submission version available at Shaw et al. (2023a). The full data used to produce all the figures and data within this paper is available at Shaw et al. (2023b).

*Author contributions.* DRS conceptualisation, data curation, formal analysis, investigation, methodology, software, validation, visualisation, writing - original draft preparation. TJC data curation, investigation, software, methodology, writing - original draft preparation. HLD, EHS and GB validation, writing - review and editing. ECC validation. ZW methodology. NC conceptualisation, formal analysis, investigation, methodology, project administration, funding acquisition, resources, software, supervision, writing - review and editing.

*Competing interests.* The authors declare that they have no conflict of interest.

*Acknowledgements.* The development of this model has been funded by grants from the Alfred P. Sloan Foundation COMMODIAC (2018-10083) and MOCCIE2 (2019-12306) for DRS and MOCCIE3 for TJC (2020-13912). Conclusions reached or positions taken by researchers or other grantees represent the views of the grantees themselves and not those of the Alfred P. Sloan Foundation or its trustees, officers, or staff. The authors would like to thank Tara Kahan and Shan Zhou for providing the raw data from Zhou et al. (2020) for comparative analysis. We would also like to thank Roberto Sommariva for his feedback during model development. For the purpose of open access a
Creative Commons Attribution (CC BY) licence is applied to any Author Accepted Manuscript version arising from this submission.



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
