# Peer review of "INCHEM-Py v1.2: A community box model for indoor air chemistry"

_EGUsphere, 2023_

## Author Response (AR1)

Below is a detailed response to the reviews of our manuscript, we have split the comments up by reveiwer, as per the instructions. We appreciate the detailed and thorough reviews that we believe have strengthened our model description manuscript. We have responded to both reviewers individually below. The response follows the format (1) reviewers comment, (2) our response and (3) the changes within the manuscript. The reviews have resulted in substantial manuscript improvements which can be seen in a revised document.

Reviewer 1

**General Comments**

1. In this model description paper, Shaw et al. present the INCHEM-Py box model. My interpretation of the paper's purpose is to provide a user of the model with a reference for the key aspects of the model: processes, parameters, assumptions, verification and examples of use. The novelty and need for such a model is without question. The paper reads very well. However, I cannot currently recommend publication without major revisions as some basic aspects of a GMD description paper have not been met. Below I explain these revisions alongside suggestions for minor and technical revisions. The most significant revision relates to the difference (as defined by GMD) between verification and evaluation. Some key aspects of the model are evaluated against observations, however there is currently no verification to show that the mathematics is operating as expected (more detail below).
2. We thank the reviewer for their time and effort. We have addressed the specific comments below, producing some significant changes in the paper.

**Specific Comments**

1. Paragraph beginning line 38 – the justification of needing a model because comprehensive measurements is difficult or lacking is misleading – the model is only as good as the measurements used to evaluate it – and the model offers more than what new measurements could supply – hindcasts, forecasts and hypothetical scenarios. Please reconsider the relationship between  model and measurement and the resulting motivation for the model.
2. We have changed the text as follows to address the reviewer's comment.
3. "An alternate approach is to simulate these processes in a model, coupled with evaluation using experimental data. It is then possible to use the evaluated model to provide forecasts and to explore hypothetical scenarios not possible through measurements, thus providing a deeper insight into the processes of interest."

1. Line 52 – witth the motivation for this study being stated as human health – there should be more comment on how the lack of microbial representation affects the limit of model application
2. The focus of this paper is on the chemical processing that occurs indoors. We agree that microbial processes also affect health, but including them is beyond the scope of this work.

1. Introduction in general – should outdoor effect of indoor emissions be a motivation for the model? There is currentlty no mention of alternative available models and why this model is novel, and there needs to be.
2. There is no alternative model to INCHEM-Py for modelling the detailed chemistry of indoor atmospheres. Other models utilise the Master Chemical Mechanism but do not have processes or schemes for indoor air chemistry. For instance, AtChem2 is designed for outdoor chemistry and PyCHAM (based on PyBox and MANIC), is designed for chamber studies and particulates. We have now added the following text:
3. "It is a refactor and improvement of the INdoor Detailed Chemical Model (INDCM) developed by Carslaw (2007) and is the most detailed chemical model that currently exists for exploring the chemistry of indoor atmospheres. It allows the user to understand the change in concentrations of indoor species over time and the key reaction pathways that describe the formation, transformation and loss of indoor species."

1. Paragraph beginning line 71 – if INCHEM-Py can only be used with the MCM plus the 'indoor chemistry' (with user-defined supplementary reactions), this needs to be stated explicitly, since this is an important facet (e.g. users with alternative chemical schemes (that could take the place of MCM plus 'indoor chemistry') cannot apply them).
2. It is possible for users to adopt an alternate mechanism, however it would need to be in the same format. Text has been added to the paper to clarify this point.
3. "The model also allows users to add their own chemical mechanisms to supplement those already included, or to use a mechanism other than the MCM (although any new mechanisms need to be in the same format as that adopted in INCHEM-Py). Such flexibility will facilitate reaction mechanism development, and community-driven model development."

1. Line 81 – here please reference the reader to the section of the paper detailing outdoor concentration representation
2. Reference added to the text.

1. Line 82 – how is the assumption of irreversible deposition justified, particularly in light of observations of reversible partitioning of some chemicals with surfaces: doi.org/10.1021/acs.est.0c00966
2. The reviewer is correct that there is some literature detailing such processes now becoming available, such as the one they highlight for SVOCs. In general though, there is insufficient information with which to parameterise such processes at present given they can vary according to the depositing gas properties, the surface in question and its condition. We do consider surface interactions for the oxidants hydrogen peroxide and ozone (see Carter et al., 2023), and could expand this implementation as more information becomes available.

1. Line 85 – I got lost with the reference to the 'new surface mechanisms', as it isn't clear what this is.
2. We agree this sentence is confusing, and have edited it to:
3. "Additional emissions, both constant and intermittent, can be user defined."

1. Line 87 – Particle reactions need to be introduced earlier in this paragraph prior to their mentioning in the total number of reactions.
2. The following line has been added to the text introducing the existence of the particle reactions.
3. "Gas-to-particle partitioning reactions are included for limonene, a-pinene and b-pinene (discussed in Sect. 2.4)."

1. Line 90 and around – Please mention how stoichiometries other than 1 are dealt with.
2. The following has been added to the text:
3. "Stoichiometries are dealt with in the MCM export by repeating the species within the reaction, i.e. the photolysis reaction $H_2O_2$ = 2OH is represented as $H_2O_2$ = OH + OH. This method is used throughout INCHEM-Py."

1. Line 99 – I am confused about how gas-particle partitioning has the same form as eq. 2 (i.e. the implication is that partitioning is reaction-based, is this correct?), please expand.
2. We agree that this sentence is confusing and we have now expanded the text to:
3. "Each species with the potential to partition to the particle-phase has a corresponding term that tracks the change in its particle-phase concentration over time. This change in concentration for each partitioning species is determined by the equilibrium between the absorption to and desorption from the particle bulk based on its individual properties (Carslaw 2007). Therefore, the particle-phase concentration of each species is represented by its own ODE in the system and individual species contributions to total particle concentration can be output (particles are discussed in detail in Section 2.4)."

1. Line 106 – please make clearer whether the Jacobian is calculated by INCHEM-Py code or the code of the ODE solver (or something else)
2. The Jacobian is calculated by INCHEM-Py code, this has been clarified in the text with the addition below. We have also removed the equation for the Jacobian based on comments from the other reviewer.
3. "LSODA makes use of the Jacobian of the system to ease concentration predictions and is calculated in full as an array of partial differential equations by INCHEM-Py."

1. Text preceding eq. (8) – it is not clear what the distinction/relationship between the Hayman, Saunders and Jenkin references are here, and therefore unclear what the provencance of l, m and n is.
2. Hayman (1997) developed a radiative transfer model used for the one day clear sky actinic flux calculations for central Europe in the summer, which is used to derive the J value parameterisations, given by Saunders (2003). These calculations use quantum cross-sections and yields from Jenkins (1997). The two paragraphs have been combined and reworded as:
3. "The solar zenith angle is used to calculate the solar photolysis portion of the overall photolysis rate (J), given by Eq. (7), for the 43 photolysis reaction rates currently included within the model. A description of the scattering method employed for this calculation is taken from Hayman (1997) and utilised in Saunders et al. (2003), where parameters l, m and n are optimised as per the discussion in Jenkin et al. (1997). These values are then attenuated according to the glass type chosen by the user, through the use of a transmission factor ($\Psi$). Indoor photolysis rate values ($\phi$) are available for seven light types, and these are summed with the attenuated outdoor values to give the total photolysis rate as shown in Eq. (7)."

1. Line 131 – why is the transmission factor weighted by photolysis properties of species? Isn't the transmission factor just a wavelength-dependent fraction of the penetrating light?
2. Yes, the transmission factor is just a wavelength-dependent fraction of penetrating light. The weighted transmission factor is then the transmission factor weighted for the wavelengths that are important for the individual photolysing species through the window. For instance, even though a glass may transmit light between 300-400nm, an individual species might only absorb light from 340-360 nm and our system accounts for that, as described in Wang et al. (2022). We have now modified the text to read as follows:
3. "The absorption cross-section and quantum yields for each photolysing species were used to calculate the weighted transmission factor, $\psi$, for each wavelength range as described in detail in Wang et al. (2022). The weighted transmission factor accounts for the proportion of the wavelength range transmitted by a particular glass type that is absorbed by each individual molecule."

1. Line 133 – are the quantum yields and cross-sections of Kowal et al. (2017) consistent with the values used in for indoor photolysis from natural light? If not, how is this justified?
2. We are not quite sure what the reviewer means by this sentence. The quantum-yields and absorption cross-sections used to calculate photolysis from artificial and natural light sources are calculated using data from IUPAC (https://iupac-aeris.ipsl.fr/) as detailed in Wang et al. (2022). Only the measured spherically-integrated photon fluxes were taken from Kowal et al. (2017).

1. Section 2.3 – please discuss if/how model users can effect light variations due to indoor settings facing different directions and having varying degrees of natural illumiation (e.g. angle of sunlight incident on windows is such that volume of room illuminated can vary).  In addition, it is not clear whether photolysis affects the outdoor concentrations of ingressing species in the same way that it affects indoor species. The comment on line 142 about outdoor OH concentrations suggest that outdoor concentrations are somehow fixed, so please clarify – i.e. are outdoor species subject to the same natural light intensity as indoor species? (By the way, I see that this is dealt with in section 2.3, therefore please reference 2.3 in the relevant section(s) of 2.2)

2. As the model has zero dimensions there is no representation of direction or location of a window. We consider that the value of the integrated photon fluxes measured 1 m from a light source are representative of an integrated average within our indoor space, as suggested by Kowal et al. (2017). Outdoor photolysis does not affect our outdoor concentrations, most of which are fixed. The only exceptions are that outdoor concentrations of NO, $NO_2$ and $O_3$ vary according to measured average diurnal profiles at specific locations, whereas OH, $HO_2$ and $CH_3O_2$ outdoor concentrations are functions which have a dependency on the solar zenith angle. In practice, the radical lifetimes are too short for these species to ingress and they are included for completeness and to support future model developments. The text the referee refers to on line 142 has now been removed and this item is dealt with in section 2.3 instead (and see next comment). The photolysis text has been reworded to read:

3. "Although the outdoor concentration of OH  varies with latitude and season, its lifetime is too short for it to ingress indoors. OH production indoors is driven by the reaction of $HO_2$ with NO in this instance, so the OH profile indoors is driven by the NO concentration. Therefore the tails of OH concentration in both panels of Fig. 3 follow the same minimum profile, only deviating with photolysis input. This minimum profile can be seen in Fig. 4(b) as the "No sunlight" trace."

1. Section 2.3 – please provide the reader with a note on how they may be limited by outdoor concentrations that are taken from European city summers – if they may be substantially limited please provide (where this is possible) a note on how they may contribute their own outdoor concentrations. Note, that I see in 2.6 the reader is told they may adjust the outdoor concentration file, but this information would be very useful in 2.3.

2. The following has been added:

3. "The outdoor concentrations used by INCHEM-Py can be adjusted to fit the requirements of the user. The default profiles of NO, $NO_2$ and $O_3$ from four European cities in the summer are provided as indicative locations with sufficient data to create the required diurnal profiles from the EEA (2018) air quality database. Further discussion on how the original profiles were created, how they may be adjusted, and how to create additional profiles is provided in the user manual."

1. Line 180 and line 157 – it is not clear exactly which species are modelled to ingress, is it just NO, NO2 and O3, or are OH, HO2 and CH3O2 also ingressing? If OH does ingress, then the statement about lifetime on line 180 appears to be conflicting.

2. Any species with an outdoor concentration will ingress. When running a default INCHEM-Py simulation, the outdoor OH contribution ($6.91 \times 10^2$ molecule $cm^{-3} s^{-1}$) is 0.009 % of the overall production rate of OH from all sources ($7.81 \times 10^6$ molecule $cm^{-3} s^{-1}$) at the peak OH concentration, so effectively negligible. Line 180 of the submitted manuscript is correct in that OH does not possess the lifetime to be able to travel far into the indoor environment before reaction. However, we agree that it is misleading in the context of a 0D model where there is some small contribution to the indoor OH concentration from outdoor OH. We have edited the line to read:

3. "…but the lifetime of OH is too short to survive ingress through a building and most OH indoors is made indoors through the above chemical reaction.
And additionally adding the following line later in the paragraph:
The influx of outdoor OH is less than 0.009 % of the total production rate of OH at 0.5 $ach^{-1}$."

1. Line 199 – If Kp is the quotient of two rates, shouldn't it be dimensionless? If I misunderstand, please provide the units of kon and koff.

2. The reviewer is correct that $k_{on}$ and $k_{off}$ are both rates with units of $m^3 ug^{-1} s^{-1}$. At equilibrium the ratio of $k_{on}$ over $k_{off}$ equals the gas-particle partitioning constant $K_p$. Therefore, during equilibrium conditions a value for $k_{off}$ can be calculated.

1. Eq. (10) – please provide a reference for this equation

2. This is now Eq. (8). Taken from Eq. (15) of Pankow (1994) and is in Sec. (2.2) of Carslaw (2012). These are referenced within the manuscript as the representation of partitioning by this method is not new to this publication or model (Leungsakul et al. 2005).

1. Section 2.4 – is confusing: neither here, nor in Carslaw et al. 2012 is a complete calculation for how the model estimates change to species gas- and particle-phase concentrations due to gas-particle partitioning – a complete calculation would show how the change is calculated in molecules/m3 (or molecules/m3/s if solved in ODE form) (the concentration unit used by the model). In addition – how often is this calculation made in the model and where (i.e. alongside chemical reaction ODEs in the solver, or separately)? Finally, please explain why Kp is called partitioning coefficient here but is called partitioning constant in Pankow (1994).

2. As in the previous comment, we have reidentified Kp as the partitioning constant, as in Pankow (1994). We have also added further detail as to how the model uses these equations when calculating the particle and gas phase concentrations. This section has been fully rewritten so please see the updated manuscript.

1. Section 2.8 – is very focussed on oxidant deposition and the resulting yields of organics. However, the reader would benefit from knowing whether organics also deposit to surfaces, and from an indication of whether it is this simulated deposition of organics that provides the organic reactant that generates aldehyde emission from surfaces – or is it the fabric of the surface itself that reacts with oxidants, (or something else)?

2. Within the model we do have irreversible deposition of organics, resulting in 3371 depositing species. However, it is not these deposited organics that provide the material that is later released following oxidant deposition. The latter process is driven by oxidant interaction with the material itself, as described in Carter et al. (2022). We have now explicitly stated that the irreversible deposition from v1.1 is retained alongside the new oxidant deposition mechanism.

3. "…These are retained in v1.2 but an additional surface deposition mechanism…"

1. Section 2 – what is the benefit and limitation of using a box model for indoor chemistry – compared to other types of model? What is the time interval used for solving ODE equations over and can this be defined by the user?

2. The following paragraph has been added in the conclusion to discuss the limitations of using a box model for indoor air chemistry:

3. "There are generally two types of models used to solve indoor air chemistry; box models such as INCHEM-Py that have no spatial dimensions but very complex chemical mechanisms (> 6000 species, > 19,000 reactions); and computational fluid dynamics (CFD) models that solve the evolution of very few species (< 20 species, < 15 reactions), but can track them spatially (Lakey et al. 2019). Neither model is able to fully represent the complexities of the indoor environment. The main assumption of box models is that the atmosphere is well mixed, and that all species are available to react with each other at all times. However, CFD models simply cannot capture the complex chemistry of the indoor environment due to computational constraints. Each model type attempts to fill a different knowledge gap. It is up to the user to appropriately define the parameters of the models to gain the most effective insight into the processes occurring indoors."
Details of the ODE solver has been added in section 2.1 as:
"The time step between outputs is set by the user in the settings file. The integrator will predict and output the solution to the ODE system at each time step. Default integrator parameters are given in the user manual."

1. Section 3.1 – please remind the reader in this section what the relevant difference between INCHEM-Py and INDCM is. Please expand on why spatial represenation is important when comparing against observation.

2. Based on comments from the other reviewer, we have removed the 4-OPA skin production comparison from the manuscript. There is discussion around spatial

representation in indoor air models is included in the conclusion. The following line has been added to section 3.2 to distinguish the INDCM from INCHEM-Py..

3. "The INDCM had less complete representations of surface interactions, indoor photolysis and indoor-outdoor exchange."

Overall

1. It is unclear what makes this a community model rather than a model.
2. The aim of INCHEM-Py is to be developed for and by the community, as opposed to the INDCM or other models that use proprietary software or require permission from the licence holder to use or modify. We have included it in the title as it is important to us that users see this as a model they can contribute to.

1. Needs some indication of processing time (and processor used) for representative runs
2. A section on the practical details of the model has been added to the paper, Sec. 4.

1. What is the benefit (and any limitation) of using Python?
2. The majority of the heavy lifting of the model, the integration, is achieved using a Python wrapped Fortran solver, so we do not believe there are significant limitations to the model imposed by Python. The model does not run slowly, well within an acceptable time frame (<20 minutes, see section 4). The main benefit of Python is its accessibility as a language, opening up the model to more users and potential contributors.

1. Where does the user access the model?
2. A section on the practical details of the model has been added to the paper, Sec. 4.

1. A figure demonstrating the code workflow would be helpful so that users know the general structure and order of processing in the model, e.g., which processes are solved inside the ODE solver – and which outside (e.g. gas-particle partitioning is based on equilibrium, so is this solved separately to the ODEs?).
2. A schematic has been added as figure 1.

1. A section in which the core of the model (gas-phase chemistry) is verified against a benchmark like AtChem2 is needed.
2. A comparison of gas phase chemistry with AtChem2 has been included, Sec. 3.1.

1. Currently missing this part of a description paper (https://www.geoscientific-model-development.net/about/manuscript_types.html#item 1): The model webpage URL, the hardware and software requirements and the license information should be given in the text. If papers are describing subsequent development to a paper already published in GMD, authors should request them to be electronically linked to the previous version(s) in a special issue, and an overview webpage will be created.
2. A section on the practical details of the model has been added to the paper, Sec. 4.

**Technical Corrections**

[All technical corrections have been addressed in the manuscript, some additional comments in square brackets below.]

Line 29, comma between windows and doors

Line 63 and elsewhere – Shouldn't letter C should be in math font to be consistent with the equation?

Sections around all equations – please consistently use math format for letters used in equations.

Equation (1) – to be consistent with the use of sigma and j for reactions, surface losses should be summed oved multiple potential surfaces

Line 68 and elsewhere – should 'air change rate' read 'air exchange rate' to be consistent with indoor research convention? Perhaps the convention has changed recently (in which case please state in rebuttal), but I notice that exchange is used in https://doi.org/10.1021/es301350x

[This is a contentious point. We have only recently begun using "air change rate" after discussion with colleagues at NIST, who argue that 'exchange' implies that the air is moving a plug flow motion and is completely exchanged.We therefore prefer to use "air change rate".]

Line 90 – when referrring to something in the gas phase, shouldn't a hyphen be used: gas-phase reactions rather than gas phase reactions?

Line 132 – full stop missing

Line 177 – I think that loosely and tightly should be swapped to fit the stated air exchange rates

Line 178 – affects rather than effects

Line 182 – space needed after comma

Table 1 – should there be /s included in units?

Line 329 – a rogue '8.'

Line 361 – 'an' change to 'a'

Line 669 - doi of Shaw 2023b looks funny

1. General Comments

   The manuscript authored by Shaw et al. describes an advanced indoor air quality model, which shows improved performance over the previous version(s) in matching experimental results. The manuscript reads well. The cited literature is proper and accredited. The mathematical expressions are correctly defined and used. The model is flexible, and future users are expected to be able to reproduce the results discussed with enough familiarity with the framework. Nevertheless, I cannot recommend its publication unless the manuscript is improved with some major revisions. Some assumptions, especially those associated with airborne particles, are not clearly outlined. The manuscript involves insufficient discussions on how the interaction between essential sources and sinks leads to model predictions per its mathematical framework. There is no information on the errors related to the experimental data used to evaluate the predictions, and the claims on the acceptability of model-experiment comparison results are not substantiated enough. While several limitations (see my specific comments outlined below) arise from the model's assumptions and parameterization, the authors are missing a thoughtful discussion on model limitations. Finally, the manuscript lacks sufficient elaboration on some equations and procedures (see my specific comments outlined below). For a model with a stand-alone user manual, it is not straightforward to draw a line for the limit of details expressed within the manuscript versus the information obtained via consulting the manual. In my opinion, the current version of the manuscript is over-referring the reader to the manual and other information uploaded into the online repository.

2. We would like to thank the reviewer for thoroughly reviewing our manuscript. We have made some significant changes to the paper based on this review and hope that we have addressed all of the specific concerns raised. We have discussed the airborne particle formation and calculation in more detail, including a discussion on limitations that we have added to all appropriate methods chapters. We believe the review has significantly improved the paper and thank the reviewer for their help in achieving this..

   The balance between referral to the manual and excessive length of the paper is a tricky one to manage, especially when the manual is also part of the submission, as is the case for a GMD model description paper. We accept that in some cases we did not expand or elaborate sufficiently on the information we have provided within the paper. We hope the edits we have made have balanced this reliance on the user manual with the detail given in the paper.

   Specific comments have been addressed below.

Specific Comments

1. Line 34: The statement "SOA are ultrafine particles..." is inaccurate. As an OA component, SOA is a fraction of PM with various sizes, not necessarily ultra-fine and breathable particles.
2. We have adjusted the sentence to say:
3. "For example, formaldehyde is a known carcinogen, and SOA are associated with a range of cardiovascular and respiratory conditions...."

1. The paragraph starting with line 54: It will be more helpful to rationalize why you consider the processes included in the model framework essential in contrast with those not considered.
2. INCHEM-Py is a continuously developing piece of software that incorporates as much of the current knowledge of indoor air chemistry as we can pragmatically include, given knowledge, time and funding constraints. The processes that we do include have been chosen based on research requirements during the development process of the INDCM and INCHEM-Py. We have added the following text:
3. "Processes included within INCHEM-Py have been developed and implemented as our understanding of the indoor environment has improved, when experimental data have supported new developments, and according to the research questions being asked. The model is continuously being improved. This paper gives a snapshot of the current version of the model, while also defining the foundation upon which future model developments will build."

1. The paragraph starting with line 65: You chose to mention the units for the Equation's left-hand side on the previous paragraph. I would also suggest mentioning units for the parameters on the right-hand side too.
2. Units for the parameters have been included.

1. Line 86: This is almost the first time you mention the particles. I suggest more clarification. Have you considered different sizes of particles? Does your model consider particle sources and sinks like dust in-tracking, deposition, and resuspension? If not, how can user-defined dynamics be integrated into the model?
2. Based on a comment from the other reviewer, we have linked the particle section and added more detail in that section. The model only considers absorption, desorption and irreversible deposition, for particles made through limonene, α-pinene and β-pinene oxidation. Expanding and improving the particle mechanism is one of the priorities for future development of INCHEM-Py, but will require a significant body of work.  It would be possible for a user to utilise the custom reactions function of the model to implement particle dynamics into the model. Alternatively the chamber box model PyCHAM (O'Meara et al. 2021) has a significantly more developed system of modelling atmospheric particles using a physics based approach and could be used for such purposes.

1. Equation (3): Why only first-order reactions? Are first-order kinetics accurate per the MCM mechanisms? Nevertheless, the equation format is confusing. I am unsure how the summation is expanded in practice. For instance, if two reactants are A and B, would the reaction rate be given by k*(CA + CB) or kA*CA + kB*CB?
2. This confusion is an error in the paper, the sigma (summation) should be a pi (product), brackets have also been added for clarity. Thus, in your example with reactants A and B, the reaction rate would be $C_A C_B k$.

1. Equation (4): The Rij could be positive or negative depending on whether species i is a reactant or a product. This distinction is not reflected in Equation (3). That expression, as the sum of non-negative multiplications, seems to be always positive.
2. A "±" has been added to the equation and prior to R on in the previous sentence.

1. Equation (5): This is unnecessarily detailed. Why would you mention the Jacobian equation when you are not providing more context on how it is utilized in the computation framework? Equation (5) is just the mathematical definition of the Jacobian. Assuming the reader needs this level of information, one expects to receive more mathematical details. If you choose to provide a high-level summary of the solution process, there is no need to post an equation for the Jacobian.
2. The equation for the Jacobian has been removed.

1. Equation (6): Explain the coefficients and constants used in Equation (6). I can only understand the logic behind the numbers -23.45 and 365.25. Note that you have made many explanations for clearer equations in previous sections.
2. Values in the equation that is now equation (5) have been explained as:
3. "...-23.45 (°) is the solar declination angle at the winter solstice, 360 (°) is a complete angle, 365.25 is the number of days in the year averaged over 4 years with a leap year, and 10 is the approximate number of days between the winter solstice and January 1st."

1. The discussion around Figure 2: Box models are prone to severe errors upon characterizing indoor photochemical phenomena. The OH concentrations featured in Figure 2 are more reconcilable with near-window areas instead of bulk indoor space (See Lakey et al., https://doi.org/10.1038/s42004-021-00548-5). I can hardly relate to indoor OH concentrations beyond $10^5$ molecules per cubic centimeters from the order of magnitude point of view. Since you have dedicated a section to photochemistry, I think the lack of dimensionality and its implications on photochemistry need to be further discussed. This comment exemplifies my broader point above that the manuscript lacks a thoughtful discussion on model limitations and critical assumptions.

2. This is now Fig. 3. We were collaborators on the Lakey et al. (2021) paper, and used INCHEM-Py to predict the reaction rates, determine the important reaction pathways and output species lifetimes as part of the analysis. We accept that with a 0D model, there are limitations in the representation of spatial variation. However, we can set the appropriate parameters within the model to still gain useful insight. We have added a section in the conclusion, based on a comment from the other reviewer, discussing this as a general limitation of box models.

   In terms of the OH representation in Fig. 3, we have stated that the simulations are using the default values of the model unless otherwise specified. In this case the room is exposed to attenuated solar radiation and indoor lighting. OH concentrations are typically reported in /cm3, as the values are so small in e.g. mixing ratios, but we have now added a comment to the figure caption to provide the equivalent values in ppt. Our calculated photolysis rates are based on photon fluxes at 1 m from light sources and can be assumed to represent a room average (see Kowal et al., 2017 and Wang et al., 2022). In the future, we may be able to adjust the photolysis based on the surface area to volume ratio of windows and their orientation.

1. Lines 143 and 144: I can hardly reconcile your interpretations for the two figure panels. The inter-seasonal gap between the peaks in Figure 2(b) is larger than the inter-latitude gap in Figure 2(a). However, you discuss differences in the latter while commenting on the former as "comparable."
2. We have added further analysis of the differences in OH concentration with latitude and season, this is incorporated in additional text in response to the other reviewer. The full text included is:
3. "Although the outdoor concentration of OH varies with latitude and season, its lifetime is too short for it to ingress indoors. OH production indoors is driven by the reaction of $HO_2$ with NO in this instance, so the OH profile indoors is driven by the NO concentration. Therefore the tails of OH concentration in both panels of Fig. 3 follow the same minimum profile, only deviating with photolysis input. This minimum profile can be seen in Fig. 4(b) as the "No sunlight" trace."

1. Lines 163 and 164: These cities do not represent well the latitude range in your model.
2. These locations were chosen as we could get the appropriate hourly data for these stations for PM2.5, $O_3$, NO, and $NO_2$ to make the profiles and are intended to be user adjusted. In future, a further expansion of the model could be to develop a database of outdoor profiles for various locations and seasons. We have added the following text in response to a similar point by the other reviewer:
3. "The outdoor concentrations used by INCHEM-Py can be adjusted to fit the requirements of the user. The default profiles of NO, $NO_2$ and $O_3$ from four European cities in the summer are provided as indicative locations with sufficient data to create the required diurnal profiles from the EEA (2018) air quality database. Further

discussion on how the original profiles were created, how they may be adjusted, and how to create additional profiles is provided in the user manual."

1. Lines 176 and 177: The air change rate does not only depend on airtightness. Even airtight buildings can have air change rates as high as 5 per hour upon window opening or operating high-speed ventilation fans.
2. We have added the following text:
3. "...and any intentional ventilation."

1. Lines 180 to 192: These two paragraphs would be more informative if they followed an alternative organization. I would suggest first mentioning the reactions. Then, argue how ACR changes will contribute to OH increase or decrease per the competing phenomena associated with reactions. Then, comment on how their interplay leads to the ACR effect per the extreme case of 0.2 and higher values.
2. This paragraph has been reorganised, as per the suggestion.

1. Lines 185 and 186: You should also mention that enhanced ACR contributed to removing OH from the indoor domain. However, the elevated production effect seems more intensified.
2. The reviewer is correct, so we have added the following text for clarification:
3. "Increasing ACR will also increase the loss rate of OH to outdoors, however, due to its reactivity this is less than 0.0025 % of the total loss rate of OH at 0.5 $ach^{-1}$."

1. General comment on Section 2.2 and Section 2.3: It is excellent to evaluate the model against published experimental data. However, the reader of modeling research also wants to see information on model verification. What does the model tell you about interactions between rate processes? What are the fundamental mechanisms driving concentrations for typical conditions? Does the mathematical behavior of model predictions follow the established physical principles? For instance, how would you interpret the local minima and maxima in Figure 4?
2. We have now addressed model verification in section 3, including a new section comparing the gas-phase photochemistry with AtChem2. In Fig. 5(a) (previously Fig. 4(a)) the concentration of OH is driven by reactions indoors which are linked to both photolysis and outdoor concentrations of reactants: the important production reactions have now been discussed. The minima and maxima shown in figure 5(a) are driven by changes in outdoor concentrations and photolysis. In Fig. 5(b), the indoor NO is driven mainly by outdoor NO, following the outdoor profile. How different parameters impact the variation of indoor species in models has been discussed by Kruza et al. (2021) (https://doi.org/10.1016/j.atmosenv.2021.118625), who carried out a Monte Carol analysis to better understand how the different input parameters affected the model output. This work showed that for the previous version of our

model (INDCM), photolysis, air change (including outdoor concentrations) and surface deposition were the primary controlling factors for most species.

1. Line 194: Why only these species? You have species more prone to sorption to particles like octanal in Table A.1. The parameterization does not seem to limit you in extending the list of partitioning species, as the parameters in Equation (10) are available for many other chemicals.
2. Apologies, we could have been clearer and have now modified the text as below. Expansion of the gas-to-particle partitioning is a priority for model development.
3. "INCHEM-Py includes gas-to-particle partitioning for the oxidation schemes for limonene, a-pinene and b-pinene, producing 610 species that partition between the gas- and aerosol-phases."

1. Equations (9) and (10): As you have chosen to present some equations from Pankow (1994), the set needs to be logically comprehensive. You need to provide expressions from kon and koff (at least within your appendices) to guide the reader on how Equation (10) is deduced from Equation (9).
2. This section has been fully reworked, taking comments from both reviewers.

1. General comment section 2.4: Most of the discussion in this section is dependent on particle size. It seems that this factor was not included in your framework. If not, this assumption must be explicitly mentioned and discussed.
2. We have now added a clarification:
3. "Carslaw et al. (2012) provide a full description of the method used to calculate the particle partitioning parameters, which is based on absorptive partitioning from Pankow (1994), whereby the phase of the species is determined by thermodynamic equilibrium. We assume that the particles are all in the PM2.5 range and focus only on chemical composition, with no representation of how these particles evolve dynamically over time."

1. Figure 5: I am confused by the results. Regardless of how the temperature is evaluated, the model should perform similarly for a given temperature. The left panel shows that the temperature estimations are almost identical for the three methods at t~13 h. Why do the predicted OH values differ simultaneously in the right panel?
2. We have now changed the text to read as follows:
3. "Temperature is a variable that changes the rate coefficients of many of the reactions in INCHEM-Py: some will increase with temperature, while others will decrease. Although the three temperature profiles are in good agreement around the middle of the day, there is a large difference earlier on given how the three methods work. This difference in methods leads to different concentrations for some of the precursors for OH, allowing them to have a differential impact on OH chemistry later, depending on

their lifetimes. The combination of the varying reaction rates and resultant concentrations produce the final profile in Fig. 6(b)."

1. Lines 230 to 232: You are not providing your reader with enough context to compare bimolecular rate constants. I suggest calculating the decay timescales (for example, in seconds) for the rate constant to assist the reader in comparing the values. How do these reactive sinks compare with ventilation and partitioning sinks?
2. We think that we are, as all other factors are effectively equal for the two terpenes. Ventilation will remove them both at the same rate and they are assumed not to partition. The only difference is the chemical removal and we have shown the chemical rate coefficients that drive the differences. Note that the full reaction rates at each time point can be found in the data attached to the paper.

1. Lines 280 and 281: Some context, preferably listing the parametrization for each surface, is needed to help the reader understand their distinctness. For instance, glass and metal are often considered in the same category in indoor modeling. Regard it this way: if we want to consider a new surface, do we need to add another surface category, or are there guides on how your proposed suit could represent new surfaces?
2. The creation of the surface mechanism is given in Carter et al. (2023) and includes parameterisation for each surface. These values are also included in the surface module of the model itself. If a user wishes to include a new surface into the model then the methods from Carter et al. (2023) can be followed. The inclusion of distinct surfaces is driven by the available experimental data. We will also continue to add surfaces and modify the rates as more data become available.

1. Lines 288 and 289: There seem to be several reactions entangled here. Are you assuming that their yields are independent? How are you combining the yields to have a resultant one?
2. We have modified the text as follows:
3. "…a new surface deposition mechanism onto multiple surfaces has been developed for INCHEM-Py v1.2, based on the work of Kruza et al. (2017) and considering the rates of deposition and secondary pollutant emissions following individual $O_3$ and $H_2O_2$ deposition (Carter et al. 2023). Given that the nature of such interactions is likely to be complex and currently not fully understood we consider the process occurs in 2 steps: deposition to the surface and loss of the oxidant, followed by emission of new species from that surface."

1. The discussion around Figure 8: Your model is definitely doing a better job in predicting ozone degradation compared to Kruza and Carslaw. Per comparison with experimental data, how would you define a good agreement? Usually, errors within

data variability ranges signal satisfactory performance. The experimental data shown in Fig.8 have no information on data variability (e.g., error bands or error bars). I could not find any information on experimental uncertainty when I checked the cited reference itself, although you pointed out uncertainties in line 353 without providing any quantitative information. Couldn't you find a study with repeated measurements? With no information on data precision, it is hard to accept your claim about satisfactory model-observation parity. For example, 4-OPA experimental and model data differ by a factor of two. Is that difference still within the experimental error and dismissable? Moreover, the model performance in predicting 4-OPA does not seem to trump Kruza's model. Regarding 4-OPA concentration, Figure 8 shows that both your predictions and Kruza's are biased. Yours at the beginning and theirs at some time later.

2. We have decided to remove this comparison from the manuscript. The lack of reporting of uncertainty, and a lack of repeated measurements, removes confidence in the result and therefore the comparison. The paper now includes a comparison with AtChem2 that verifies the solution to the gas-phase mechanism, and we retain the experimental comparison of bleach cleaning. We also link published works that have utilised INCHEM-Py, some for experimental comparison, in the conclusion of the paper.

1. The discussion around Figure 9: Are the outdoor concentrations used to create this figure comparable with the conditions associated with Table A.1 values?
2. This is now Fig. 10. They are the same values. The following line has been added:
3. "The surface to volume ratios were set to represent the experiment and all other values were default, including the outdoor species concentrations."

1. The text within parentheses in line 352: I suggest only referring to the manuscript sections instead of a Python file in your code repository.
2. Reference to the model file removed.

1. Lines 360 and 361: Is this difference significant enough to be discussed? You have dismissed more considerable differences between the model and observations at t~16.9 hours as cases of good model-observation agreement.
2. It might be an obvious statement that the peak is lower with concrete/wood, but we don't discuss it in detail and we don't believe it detracts from the paper. The purpose of the concrete/wood simulation is simply to show that the model can highlight differences given alternative parameters that would be time-consuming to repeat experimentally.

1. General comment on discussions around Figure 8 and Figure 9: There could be objections to this strategy of adjusting emission rates to reproduce initial

observations. There is a hidden assumption with this methodology that your model grasps all the applicable physicochemical phenomena, which is not the case. How alternative emission rates would impact your estimations? I would suggest some sensitivity analysis to assess model performance during other conditions.

2. Figure 8 has been removed and figure 9 is now figure 10.. We think this is a reasonable approach. In the absence of all of the desirable input data, it is necessary to make assumptions about some of the experimental parameters. With the peroxide emission rate, the starting point for this simulation was understanding how much cleaning product was used, and the % of peroxide within that. The emission rate was calculated from Zhou et al. (2020) based on the volume and content of the cleaning product, not simply to match the measured emission. A full Monte Carlo sensitivity analysis was carried out for the INDCM by Kruza et al. (2021), which simulated 1000 variations of input parameters to identify which parameters are important to get correct to predict representative chemistry, but such an analysis is beyond the scope of this study.

1. General comment on Appendix A: i)Some of the updates between versions 1.1 and 1.2 are noticeable (e.g., acrolein changes by more than one order of magnitude), while some are minute (e.g., the updated value for methane is different from the older estimation by less than 1%). What is your criterion for implementing an update? I believe a sensitivity analysis could be helpful. How much are your results sensitive to changes in outdoor concentrations of a species? Do these changes matter in light of the sensitivity estimates?

2. The criterion for the change in the outdoor concentration value was based on an extensive literature review. We aimed to increase the number of species in the model with a representative outdoor value, but also to update the concentrations of those species already represented through the review (more data have become available since the original version of the model). A comparison showed obvious changes in species where there was previously no outdoor source, and in cases such as acrolein, where the change is substantial (~70 % average decrease from v1.1 values). As the total levels of VOCs are also changed we see this reflected in other species such as ozone (+1.4 %), OH (+17 %), $NO_2$ (+2.4 %), NO (-1.6 %) and $HO_2$ (+14.2 %).

1. ii) Most of these references cited in the table precede in publication time v1.1 and v1.2. So, which version did they serve? v1.2? More clarification is appreciated.

2. References refer to the v1.2 concentrations, the table title has been adjusted to make this clear.

1. iii) Outdoor concentrations are pretty dependent on location and time. The references you are citing here are different from each other regarding those parameters. For instance, Uchiyama et al.'s numbers pertain to Japan, whereas Bari and Kindzierski estimations are for Calgary, Canada. It would help if you cautioned the reader about the spatial and temporal variability of outdoor changes. You also need to add another column to this table to explicitly signal the location and time attributes of the cited references to prevent the reader from misunderstanding that the values are globally applicable.
2. Table A2 has been added to show the location, period of measurement and length of study for all of the cited literature.

1. General comment on Appendix B: This text could be eliminated and transferred to the online repositories. The reader has little idea how these variables are handled in your source code. So, what is the point of putting it here as if it is just coming out of nowhere?
2. We have removed appendix B, the variables are available in the manual, the model files and are included in the output of simulations.

1. General comment for the whole manuscript: I understand that the main source code consists of several functions and sub-programs that cannot be directly discussed in the manuscript. However, it will be helpful to have a figure, at least in an appendix section, demonstrating the general workflow of operating the model. A high-level flow chart referencing the code's functions/variables will work great in that capacity.
2. A flow chart of the modules and data has been included as figure 1 in section 2.

Technical Corrections:

[We have corrected the appropriate technical corrections and have identified other corrections throughout the paper, further author comments are in square brackets.]

The complete list of typos and punctuation errors is longer than what follows. However, I guess you can easily tackle those upon another round of proofreading after implementing the revisions.

Lines 18 and 19: The cited link is invalid. You can use this one: https://www.who.int/publications/i/item/9789289002134.

Line 28: A comma is missing before "and."

Line 29: Missing commas. It should be "windows, doors, and cracks."

[We have not used the Oxford comma consistently throughout, as per the GMD English guidelines and house standards (https://www.geoscientific-model-development.net/submission.html#english)]

Line 132: Missing period after the word "range."

Figure 7: The limonene arrows seem more red than orange.

[Now figure 8. This may be an error with printer or monitor colour representation, the line has the hex code #ff7f0e]

Figure 8: Your choices of color are not effective here. Grey and black are too similar. Such a bright yellow is not suitable for a white background.

[Figure 8 has been removed, we have changed the colour scheme of the majority of the figures within the manuscript, there are now no yellow lines on a white background.]

---

## Referee Report (RR1)

The revised version of the manuscript involves remarkable improvements contributing to the paper's novel contribution, clarity, and usefulness for the broad society of GMD readers. Shaw et al. have provided thorough and convincing explanations in response to the feedback received from the first draft peer review. Some of the concerns I raised were not directly responded to, and the manuscript still has the capacity for further improvement. Nevertheless, those points are not too severe to baffle the manuscript publication. I think the revised manuscript is okay for publication in its current format.

---

## Author Response (AR2)

Author response to review for EGUSPHERE-2023-1328

We once again thank the reviewers for their time and effort in reviewing our manuscript. We have addressed the final comments below.

Technical Corrections
Line 62: repeat of 'the'
-   Corrected
Line 228: format k in kon to be same as on line 232
-   Corrected
Line 264: should 'chemical' be 'gas'?
-   We have changed this to 'gas'.

Minor Revisions
Equation 1 (line 72): the final term ktCi suggests that timed emission rates are dependent on the indoor concentrations (Ci), if this is correct, then please explain why in the main text, otherwise please correct.
-   $C_i$ has been removed from the equation.
Line 98: sounds like particle-phase chemical reaction is being solved, if this not the case, then please reword
-   There are no particle-phase reactions being solved by INCHEM-Py, we have adjusted the text to now read:
    "The total size of the mechanism solved by INCHEM-Py (v1.2), before any additional user mechanisms are added, is 6507 chemical species and particles undergoing 19581 gas-phase reactions and gas-particle partitioning reactions"